# Make-it-Real: Unleashing Large Multimodal Model for Painting 3D Objects with Realistic Materials

**Ye Fang**[*]
Fudan University
Shanghai AI Laboratory
yefang23@m.fudan.edu.cn

**Zeyi Sun**[*]
Shanghai Jiao Tong University
Shanghai AI Laboratory
szy2023@sjtu.edu.cn

**Tong Wu**[†]
The Chinese University of Hong Kong
Stanford University
wutong16@stanford.edu

**Jiaqi Wang**
Shanghai AI Laboratory
wangjiaqi@pjlab.org.cn

**Ziwei Liu**
S-Lab, NTU
ziwei.liu@ntu.edu.sg

**Gordon Wetzstein**
Stanford University
gordon.wetzstein@stanford.edu

**Dahua Lin**[†]
The Chinese University of Hong Kong
Shanghai AI Laboratory
CPII under InnoHK
dhlin@ie.cuhk.edu.hk

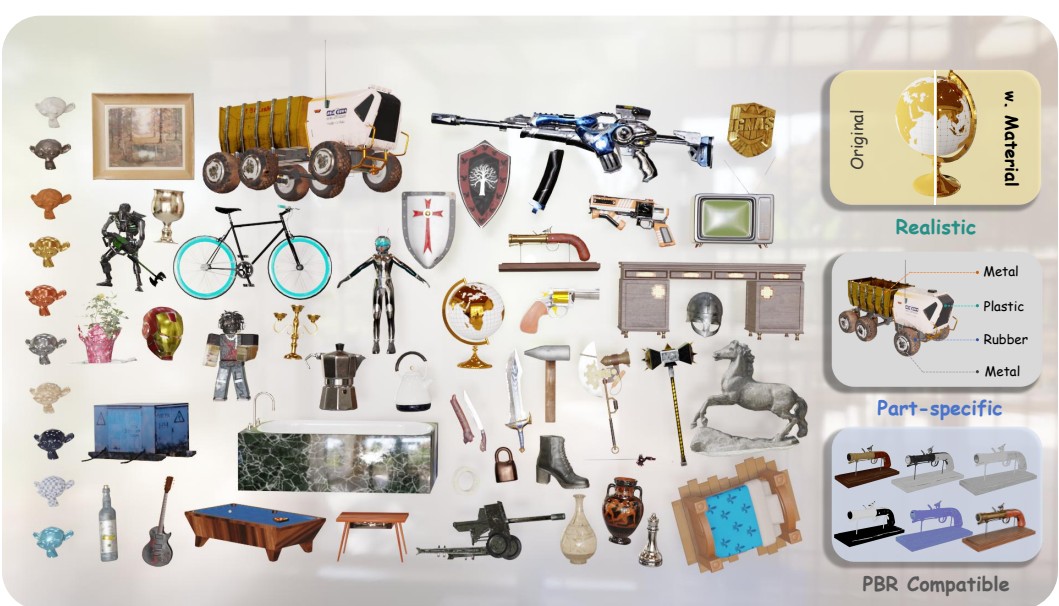

Figure 1: **Usage of Make-it-Real**. Our method can refine a wide range of albedo-map-only 3D objects from both CAD design and generative models. Our method enhances the realism of objects, enables part-specific material assignment to objects and generate PBR maps that are compatible with downstream engines.

## Abstract

Physically realistic materials are pivotal in augmenting the realism of 3D assets across various applications and lighting conditions. However, existing 3D assets and generative models often lack authentic material properties. Manual assignment of materials using graphic software is a tedious and time-consuming task. In this paper, we exploit advancements in Multimodal Large Language Models (MLLMs), particularly GPT-4V, to present a novel approach, **Make-it-Real**: **1)**

[*]Equal Contribution
[†]Corresponding Author

38th Conference on Neural Information Processing Systems (NeurIPS 2024).

We demonstrate that GPT-4V can effectively recognize and describe materials. **2)** Utilizing a combination of visual cues and hierarchical text prompts, GPT-4V precisely identifies and aligns materials with the corresponding components of 3D objects. **3)** The correctly matched materials are then meticulously applied as reference for the new SVBRDF material generation according to the original albedo map, significantly enhancing their visual authenticity. Make-it-Real offers a streamlined integration into the 3D content creation workflow, showcasing its utility as an essential tool for developers of 3D assets. Our project website is at: https://SunzeY.github.io/Make-it-Real/.

# 1 Introduction

High-quality materials are important for the nuanced inclusion of view-dependent and lighting-dependent effects in 3D assets for traditional graphics pipelines, critical for achieving realism in gaming, online product showcasing, and virtual/augmented reality. However, many existing assets and generated objects often lack realistic material properties, limiting their application in downstream tasks. Furthermore, creating hand-designed realistic textures necessitates specialized graphic software and involves a laborious and time-consuming process, compounded by significant creative challenges [32].

Traditional computer graphics methods have been either manually creating materials or reconstructing them from physical measurements. Emerging text-to-3D generative models [7, 34, 55, 10, 73, 50, 19, 51] and image-to-3D generative models [69, 58, 25, 63, 22, 62] are successful in creating complex geometries and detailed appearances, but they struggle to generate physically realistic materials, hampering their practical applicability. Recent studies have also explored advanced aspects of appearance generation [11, 75, 70, 8, 40]. However, they often rely on simplified material models. For instance, [40] lacks the ability to produce metallic maps. All these approaches do not generate corresponding displacement and height maps, which restricts the diversity and realism of the generated materials, especially regarding depth and tactile qualities. Furthermore, these methods typically require relatively long training and inference time. Considering the abundance of high-quality 3D assets online [16, 17] that lack material attributes, and the maturity of 3D generative models in geometry and albedo modeling, we aim to recover materials from high-quality geometry and base-colored 3D meshes.

However, extracting and recovering material representations for 3D meshes is challenging. Unlike previous material recognition methods [40, 4, 20], this difficulty is heightened when identifying and separating different material regions within 3D objects in constricted albedo textures. These maps only reflect the base color and can be distorted, as shown by the globe in the upper right corner of Figure 1. Additionally, shadows and lighting can affect judgment. Thus, the model must have strong material recognition capabilities and prior knowledge of object types and materials.

The emergence of Multimodal Large Language Models (MLLMs) [5, 46, 12, 2, 23, 59] provides novel approaches to problem-solving. These models have powerful visual understanding and recognition capabilities, along with a vast repository of prior knowledge, which covers the task of material estimation. Specifically, we are using GPT-4V(ision) for matching of materials. We first create highly detailed descriptions for materials to build a comprehensive library. Next, we use GPT-4V to retrieve materials for each segmented part of the object, utilizing visual prompts [72] and hierarchical text prompts. Finally, we meticulously designed algorithms to generate SVBRDF maps with consistent albedo, achieving realistic visual quality.

Notably, our work differs from the aforementioned studies by leveraging prior knowledge from foundation models like GPT-4V to extract and infer materials in albedo-only constrained scenarios. Additionally, we utilize existing material libraries as references to generate corresponding SVBRDF maps on a designed region-to-pixel algorithm. As illustrated in Figure 1, our approach features: **1)** Enhanced 3D mesh realism: Leveraging GPT-4V's visual perception and external knowledge, our method improves the realism, depth, and visual quality of a wide range of mature 3D content generation models. **2)** Part-specific material matching: Ensuring material consistency with a segmentation network and refinement process, enabling precise material property retrieval for each segment. **3)** Rendering engine compatibility: Generating six comprehensive material maps(roughness, metallic, specular, normal, displacement, height), which are compatible with downstream rendering engines. Developers only need to paint albedo textures; material properties are then automatically generated, saving extensive time on detailed ambient occlusion masks and material map creation.

In summary, our contributions are as follows:

- We present the first exploration of leveraging multimodal large language models (MLLMs), e.g.,GPT-4V for material recognition and unleashing their potential in applying real-world materials to extensive 3D objects with albedo-only.

- We create a material library containing thousands of materials with highly detailed descriptions readily for MLLMs to look up and assign.

- We develop an effective pipeline for texture segmentation, material matching and SVBRDF maps generation, enabling the high-quality application of materials to 3D assets.

## 2 Related Work

**3D Object Generation.** The generation of 3D models using deep learning methods has experienced rapid development in recent times. The mainstream research can be primarily divided into two categories. The first category relies on techniques that optimize a Neural Radiance Field (NeRF) [43] or 3D Gaussian [30] guided by 2D diffusion model through score-distillation-sampling(SDS) loss [48, 42, 39, 8, 64, 53, 56, 67]. The second category aims at obtaining 3D representation via direct inference model, e.g.,, Point-E [44], Shap-E [28], and LRM [25], proven fast with high quality through large-scale pretraining [24, 22, 57, 35, 55, 71, 58, 63, 65]. Although the capabilities of these methods are continuously improving, they still lack a high degree of realism in textures. More importantly, since the textures of 3D objects obtained by these methods are shaded, they cannot directly respond to lighting changes under different lighting conditions, leading to less realism. Although some works attempt to generate PBR textures, their results are considerably limited to generate physically realistic materials due to the low robustness of SDS [75, 8]. Our work is the first to introduce the prior knowledge of MLLMs into the texture synthesis process. We verifies that our method can be seamlessly and effectively applied to generated 3D objects, facilitating their downstream applications under different lighting condition.

**Material Capture and Generation.** Recent studies such as Material Palette [40], MatSim [20], TwoshotBRDF [4] have made progress in the recognition and extraction of 3D object materials, allowing for the retrieval of SVBRDF information from images of real materials [40] and combining object shape and illumination [4], but they fail to extract and infer the materials of 3D objects with only albedo. On the other hand, works like Paint-it [75], Matlaber [70], Collaborative [60], Fantasia3D [8], and TANGO [11] focus on generating text-controllable 3D meshes with physically-based rendering material properties. However, these methods require either extensive training time on BRDF datasets or long inference time for generating materials. Additionally, they are limited to synthesizing only a subset of PBR textures and cannot generate the full range of material properties, such as height and displacement, which are essential for the fine tactile perception of object surfaces.

**Multimodal Large Language Models.** In the wake of the advancements achieved by large language models (LLMs) [5, 46, 12, 2, 23, 59], domain of research has increasingly turned its attention towards multimodal large language models (MLLMs). Recent advances in this field focus on the integration of vision understanding capabilities with LLMs [76, 1, 37, 36, 26, 21, 3, 18, 54, 49, 6].The advent of GPT-4V [45] has marked a significant milestone in the evolution of MLLMs, demonstrating groundbreaking 2D comprehending capabilities and open-world knowledge. Although GPT-4V cannot directly process 3D data, a pioneering work , GPTEval3D [68], first exploited GPT-4V's ability in evaluating the quality of generated 3D objects, and found that GPT-4V's judgement was in line with human evaluation. In this work, we delve into a new application of GPT-4V for material assignment of 3D objects.

## 3 Methodology

### 3.1 Preliminary

Physically Based Rendering (PBR) materials are a compact representation of the bi-directional reflectance distribution function (BRDF), which describes how light is reflected from a surface. PBR material maps primarily encompass seven attributes: **A**lbedo, **R**oughness, **M**etallic, **N**ormal, **S**pecular, **H**eight, and **D**isplacement. Based on the rendering equation [29], given a location $x$ and the surface

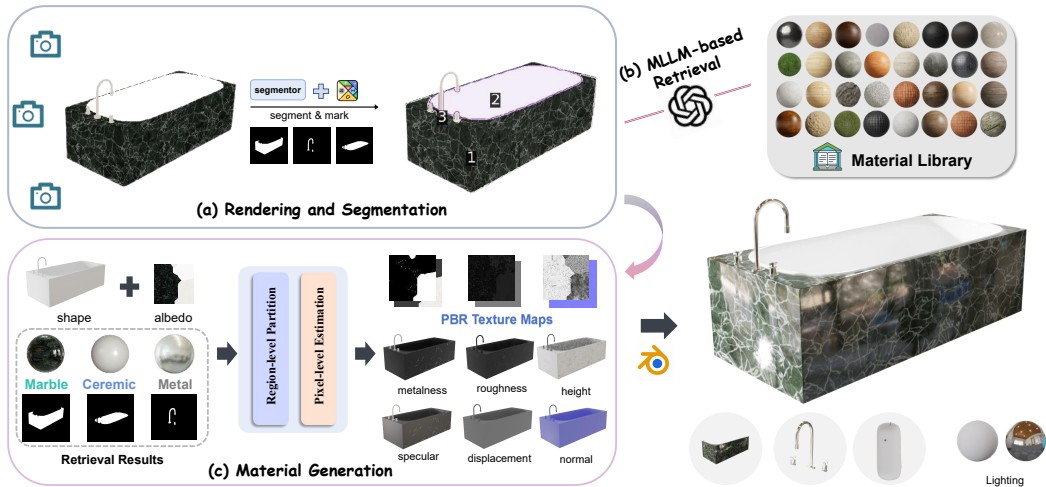

Figure 2: **Overall pipeline.** This pipeline of Make-it-Real is composed of image rendering and material segmentation, MLLM-based material retrieval, and SVBRDF Maps Generation. We finally use blender engine to conduct physically-based rendering.

normal $n$, the incident light intensity at this point is denoted as $L_i(\omega_i; x)$ along the direction $\omega_i$; BRDF $f_r(\omega_o, \omega_i; x)$ denotes the reflectance coefficient of the material viewing from direction $\omega_o$. The observed light intensity $L_o(\omega_o; x)$ is calculated over the hemisphere $\Omega = \{\omega_i : \omega_i \cdot n > 0\}$:

$$L_o(\omega_o; x) = \int_\Omega L_i(\omega_i; x) f_r(\omega_o, \omega_i; x)(\omega_i \cdot n) d\omega_i. \tag{1}$$

Given the advancements in generating high-quality 3D shapes with albedo maps, the restoration of realistic material properties remains a challenge. We highlight a novel problem: given a 3D mesh ($\tilde{O}$) and known Albedo ($\tilde{A}$) map, which reflect the object's intrinsic appearance, the goal is to extract and restore all other SVBRDF attributes of the object, i.e. *Make-it-Real*$(\tilde{O}, \tilde{A}) = \{R, M, N, S, H, D\}$. Our setup supports the popular Cook-Torrance analytical BRDF model [13]. In this parameterization, the BRDF includes components for albedo $b_a \in \mathbb{R}^3$, metallic $b_m \in \mathbb{R}$, and roughness $b_r \in \mathbb{R}$. For more complex surface simulations, such as displacement and height modeling, we use the Blender rendering engine to simulate the BRDF function $f_r(\omega_o, \omega_i; x)$.

## 3.2 Make-it-Real: A Framework for Material Matching and Generataion

In this section, we outline our material matching and generation pipeline, illustrated in Figure 2, which encompasses three stages: rendering and segmenting 3D meshes, retrieving matching materials using MLLM, and generating spatially varying BRDF maps from coarse to fine.

### 3.2.1 Rendering and Material Segmentation

To accurately segment different material regions on 3D meshes with albedo maps, we propose an innovative segmentation strategy based on 2D image rendering in Figure 2 (a). Initially, we use rasterization to render the input albedo mesh from various viewpoints to obtain a series of images:

$$\mathcal{I}(x, y) = \mathcal{R}\left(\text{UV}_{\text{map}}\left(\text{rasterize}(\tilde{O}, v_t, x, y)\right), \mathcal{T}\right). \tag{2}$$

where $\mathcal{I}(x, y)$ is the pixel value at image coordinates $(x, y)$, rasterize$(\tilde{O}, v_t, \cdot)$ maps the 3D mesh $\tilde{O}$ from viewpoint $v_t$ to 2D screen coordinates, $\text{UV}_{\text{map}}(\cdot)$ converts rasterized coordinates to UV coordinates, $\mathcal{T}$ is the albedo map, and $\mathcal{R}(\cdot, \mathcal{T})$ samples color from the $\mathcal{T}$ using the UV coordinates.

For the rendered images, we employ the Semantic-SAM [33] to perform preliminary semantic segmentation. Empirically, we select the main viewpoint with the largest projected area of the mesh, as it is more likely to contain more details. To address potential over-segmentation, drawing inspiration from [74], we extract non-overlapping segments from the masks to form distinct patches,

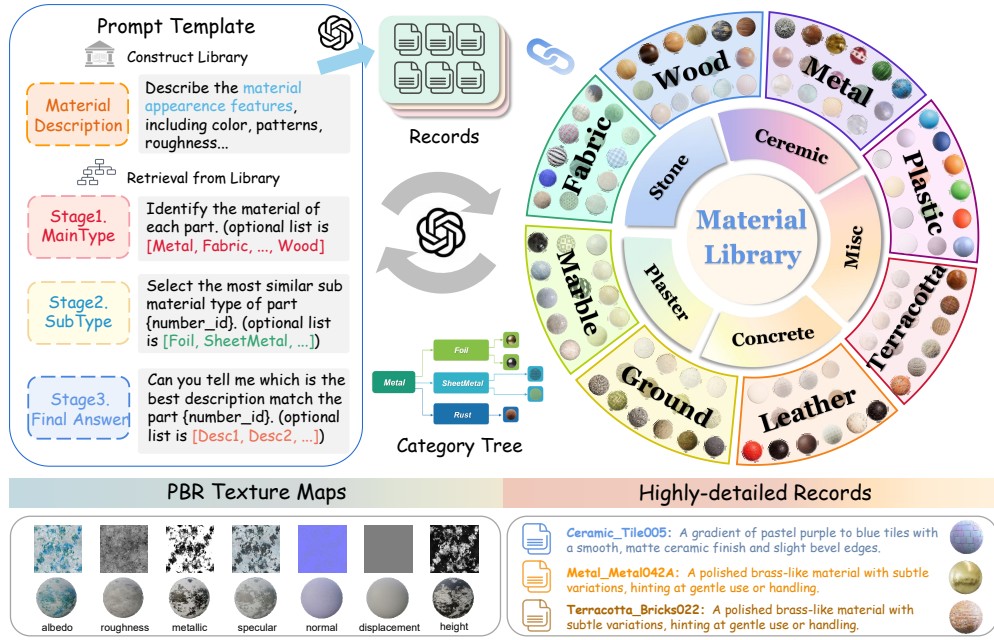

Figure 3: **The process of MLLM retrieving materials from the Material Library.** Utilizing GPT-4V model, we develop a material library, meticulously generating and cataloging comprehensive descriptions for each material. This structured repository facilitates hierarchical querying for material allocation in subsequent looking up processes.

as detailed in our approach in Figure 4 (a). These patches are then merged based on similar colors to obtain the final material grouping. We incorporate Set-of-Mark [72] method to annotate each material segment with a unique identifier, sorted by area size from largest to smallest. This annotation acts as a visual cue, enhancing the visual comprehension capabilities of large multimodal language models.

### 3.2.2 MLLM-based Material Retrieval

**Material library with fine-grained annotations.** To enable large multimodal language models to accurately retrieve and match materials, we construct a finely annotated material library, as shown in Figure 3. It is composed of three main components: comprehensive PBR texture maps, highly-detailed records, and a category tree. It comprises 1,400 unique, tileable materials spanning 13 primary categories and 80 subtypes. The data primarily derives from the [61], which offers comprehensive PBR material textures under a CC0 license with 4K resolution. Each material is represented by seven maps: albedo, roughness, metallic, specular, normal, displacement and height. Accompanying each material are highly-detailed annotations by GPT-4V, offering thorough descriptions of the material's visual characteristics and rich semantic information for the subsequent retrieval process. Created by crawling material sphere images and constructing prompts, these annotations capture subtle differences between materials, facilitating precise retrieval by GPT-4V, as detailed in Appendix B.4.

**Hierarchical prompting for material retrieval.** Due to the vast size of our material library, feeding all prompts to GPT-4V simultaneously proves inefficient and challenging for memory retention. To ensure efficient and accurate material allocation in segmented areas of 3D meshes, we adopt a hierarchical text prompting approach. The schematic of the designed prompt is shown on the left side of Figure 3, and a complete querying process unfolds in Appendix B.4. This method starts by identifying the primary material types corresponding to each labeled region. Subsequently, hierarchical prompts guide GPT-4V to distinguish specific subclasses within the main material categories. We retrieve all descriptions under these subclasses to ascertain the most fitting descriptions for the segmented blocks. This hierarchical processing enables a more granular search of our material library, identifying the optimal description for each material segment. This approach aids in assigning the most suitable materials to each region and reduces memory and time consumption.

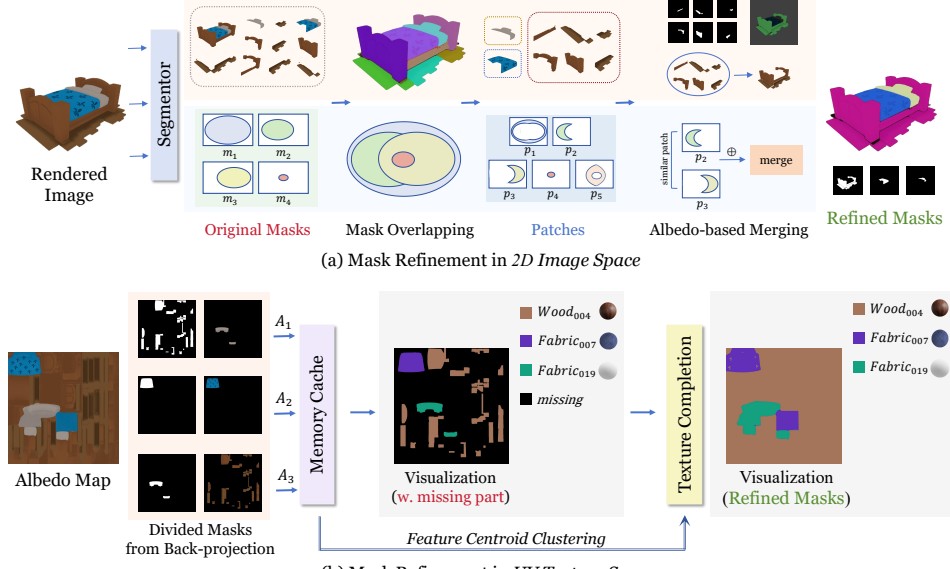

(a) Mask Refinement in *2D Image Space*

(b) Mask Refinement in *UV Texture Space*

Figure 4: **Illustrations of mask refinement in 2D image space and UV texture space. (a)** We effectively cluster concise material-aware masks compared to original segmented parts from [33]. **(b)** We fix missing parts on the uv texture space to get a complete texture partition map.

### 3.2.3 SVBRDF Maps Generation

We propose a method to generate SVBRDF maps on a region-to-pixel scale in Figure 2 (c). Initially, we segment texture map in uv space based on queried material regions on 2D image space. We then estimate BRDF values in pixel space using the object's original albedo map for reference, ensuring consistency with the albedo map. This approach effectively enhances the realism of rendered surfaces.

**Region-level texture map partitioning.**    Upon acquiring segmentation masks for material regions within the 2D rendered space of a 3D mesh, our objective transitions to transposing these segmentations from the 2D image space to the corresponding UV space. As described in Section 3.2.1,we extract 2D image features from 3D mesh points via rasterization, and then we apply the material masks to these features, facilitating the accurate transfer of segmentation to UV space. To project the image feature $I_t$ back to the texture atlas $\mathcal{T}_t$ with segmented image mask $m_t$ at the viewpoint $v_t$, we apply gradient-based optimization for $L_t$ over the values of $\mathcal{T}_t$ when rendered through the differential renderer $\mathcal{R}$, as presented in Equation (3). In Equation (4), we then compute the difference between $\mathcal{T}_t^{mask}$ and the initialized $\mathcal{T}_t$ to transfer the mask image feature $m_t$ into the texture space, represented by $m_{uv}$. The term $\sigma$ denotes the difference coefficient. Due to the limited perspectives available in rendering, we avoid using a naive median-filling approach to ensure that there are no missing areas on the texture map. Instead, we employ a block-centric clustering based on albedo, as illustrated in Figure 4 (b), to obtain cohesive and refined region masks. The process is shown in Appendix B.1

$$\nabla_{\mathcal{T}_t} L_t = (\mathcal{R}(\text{mesh}, \mathcal{T}_t, v_t) - I_t) \odot m_t \frac{\partial \mathcal{R}}{\partial \mathcal{T}_t}. \tag{3}$$

$$m_{uv} = \mathbf{1}\left( \sum_{i=1}^{3} |\mathcal{T}_t^{mask} - \mathcal{T}_t| > \sigma \right). \tag{4}$$

**Pixel-level albedo-referenced estimation.**    To achieve precise estimation of spatially varying BRDF (SVBRDF) at the pixel level, we draw inspiration from techniques commonly utilized by artists in creating texture maps. Artists typically use albedo maps as a reference for constructing ambient occlusion masks and further generating SVBRDF maps for material properties, which occupies a significant time portion of appearance modeling. Our method involves using the albedo map of the original object as a reference to refine the retrieved materials. We enhance the querying process using a KD-Tree algorithm, which searches for the nearest neighbor pixel index in the key albedo(retrieved map) for each RGB value of the queried albedo(input map) pixel, detailed in Appendix B.2. This process ensures that areas with similar colors exhibit similar BRDF values, avoiding abrupt changes

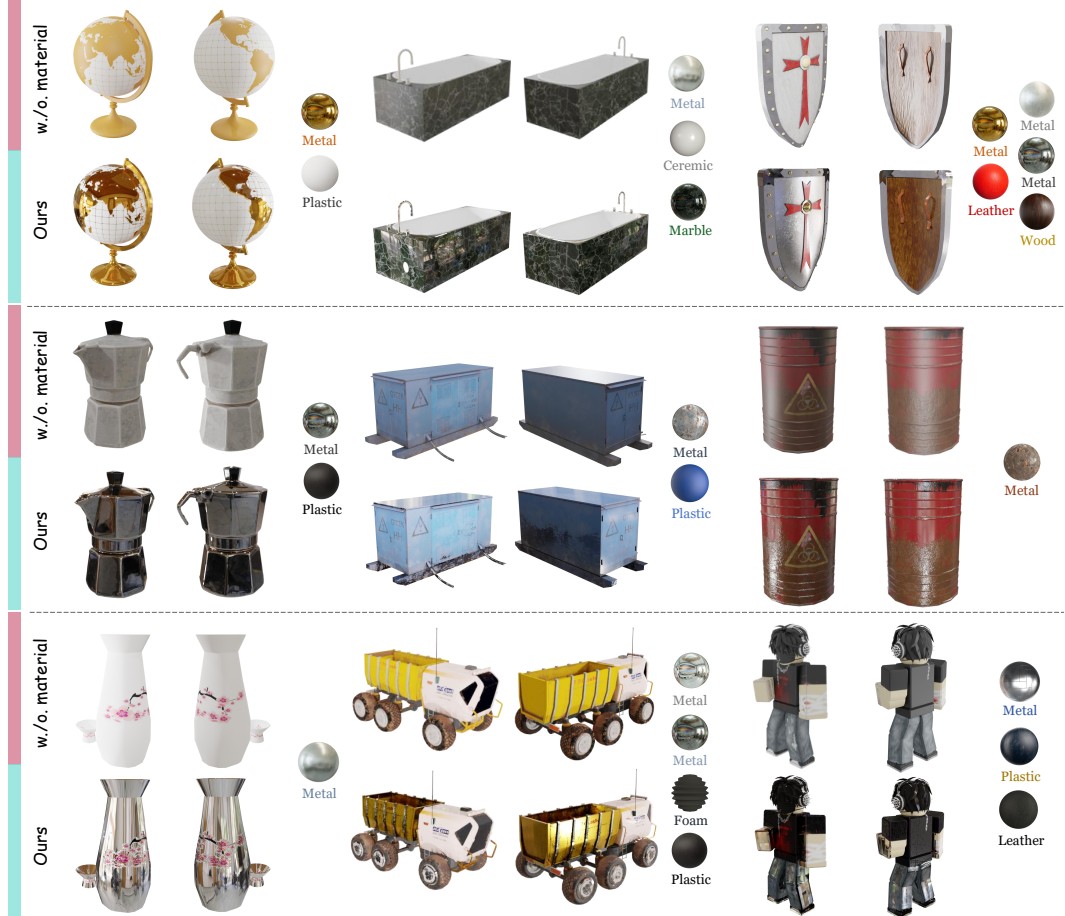

Figure 5: **Qualitative results of Make-it-Real refining 3D asserts without PBR maps**. Objects are selected from Objaverse [15] with albedo maps only.

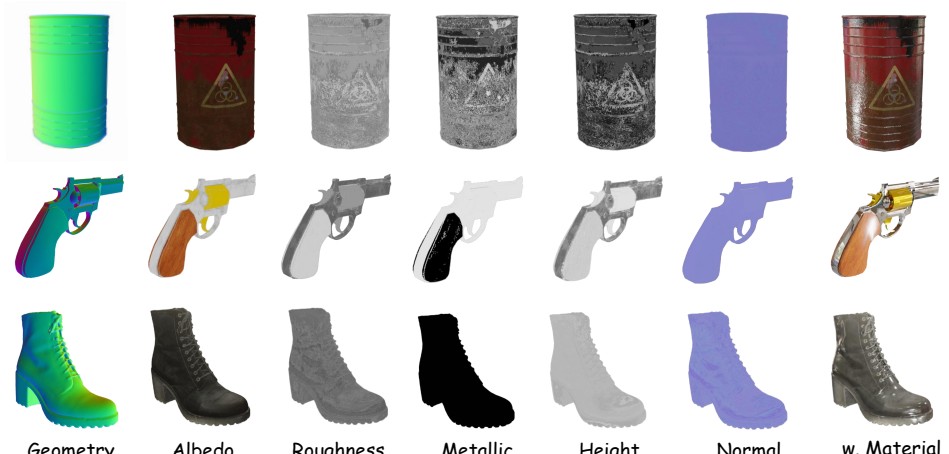

Figure 6: **Visualization of generated texture maps**. We visualize some SVBRDF maps, where the material maps are well aligned with the albedo maps.

in material properties; For regions with greater color differences, the distribution of differences aligns consistently with the input albedo, to simulate variations such as embossments or scratches. We retrieves SVBRDF values at pixel level, maintaining texture consistency with the albedo and producing appropriate concave or smooth surface effects. We further analyze the effects in Section 4.2.

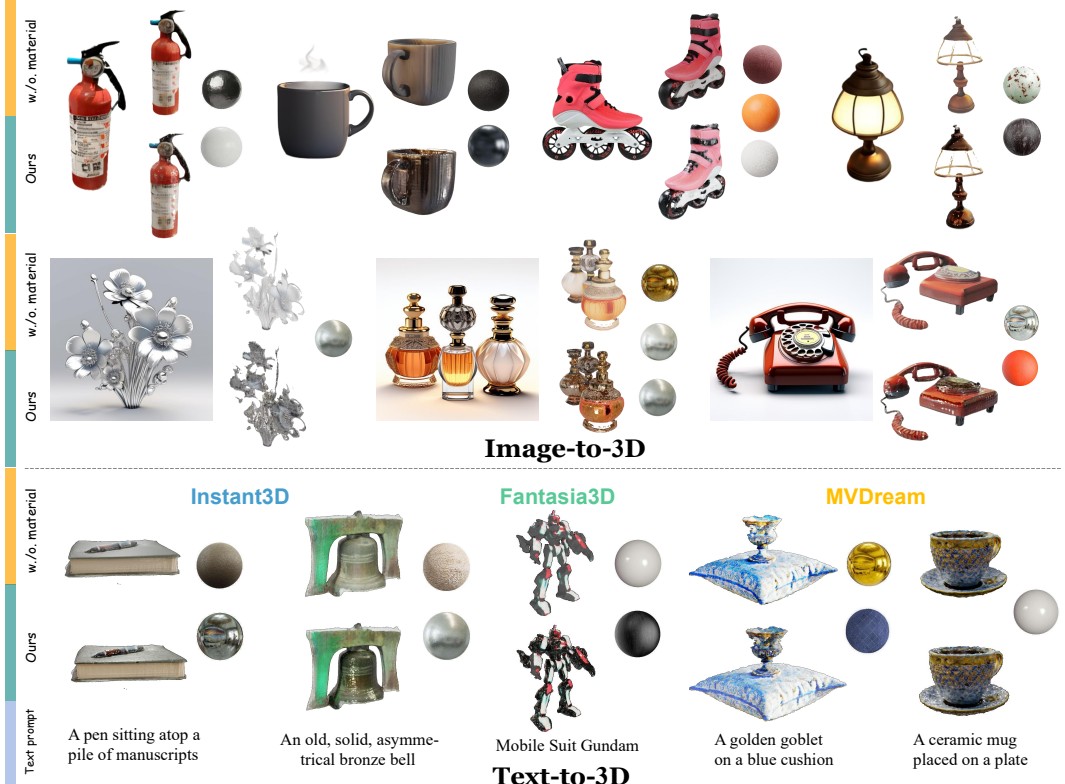

Figure 7: **Qualitative comparisons** between Make-it-Real refining results and 3D objects generated by edge-cutting 3D content creation models. The upper row depicts image-to-3D models (InstantMesh and TripoSR), and the lower row shows results of text-to-3D models.

# 4 Experiments

## 4.1 Experiment settings

To verify the effectiveness of Make-it-Real, we conduct refinement experiments mainly on two types of objects. The first type is artificial 3D assets, with the primary model from Objaverse [15] filtered by [55]; The second type is objects generated by state-of-the-art 3D generation methods. For existing 3D assets, we pick 200 objects with diverse textures from Objaverse by human experts. For 3D generative models (InstantMesh [69], TripoSR [58], MVDream [53], Instant3D [34] and Fantasia3D [8]), we also generate 200 objects for each methods use prompts designed in GPTEval3D [68]. We use Make-it-Real to refine the objects and compare the texture quality before and after refinement. We perform both GPT-4V [45] based evaluation and user study on the above objects (Detailed guidance and prompts for evaluation are available in Appendix B.5).

In addition to evaluation details, we also provide information about the rendering procedure in Appendix B.3 for reproducibility. This includes the rendering tools used in the experiments, hyper parameters related to back-projection, and the basic performance metrics of the model during the experiments, such as time, memory, and number of queries.

## 4.2 Experiment results

**Texture refinement for existing 3D assets.** As shown in Figure 5, assets processed through Make-it-Real demonstrate the capability to accurately segment objects, assign various suitable materials, and synthesize high-fidelity, photorealistic textures. Some materials exhibit notable highlights, such as a marble bathtub and a globe made of gold. The interaction of these different materials with light varies significantly, leading to diverse reflective effects under the same environmental lighting, thus presenting a range of textures. Additionally, material properties vary across different regions; for instance, the globe's landmass and handle exhibit gold characteristics, while other parts are identified

Table 1: **GPT evaluation and user preference**. GPT's and user's preference comparison on Make-it-Real refined objects sourced from existing 3D assets and state-of-the-art 3D generation methods.

| Domain | Method / Source | GPT Evaluation | | User Preference | |
|---|---|---|---|---|---|
| | | Base object | +Make-it-Real | Base object | +Make-it-Real |
| 3D assets | Objaverse [15] | 15.2% | 84.8% | 22.2% | 77.8% |
| Image-to-3D | InstantMesh [69] | 28.2% | 71.8% | 31.1% | 68.9% |
| | TripoSR [58] | 36.4% | 63.6% | 33.0% | 77.0% |
| Text-to-3D | Instant3D [35] | 38.5% | 61.5% | 35.4% | 64.6% |
| | MVDream [53] | 44.1% | 55.9% | 41.5% | 58.5% |
| | Fantasia3D [8] | 46.2% | 53.8% | 48.7% | 51.3% |

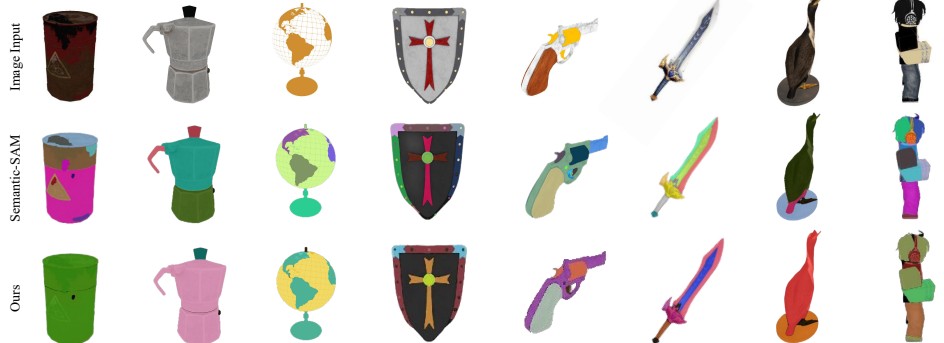

Figure 8: **Ablation study of material segmentation refinement.** Compared to direct usage of SemanticSAM [33], Our post-process tailored for material segmentation on 3D object can produce more consistent results.

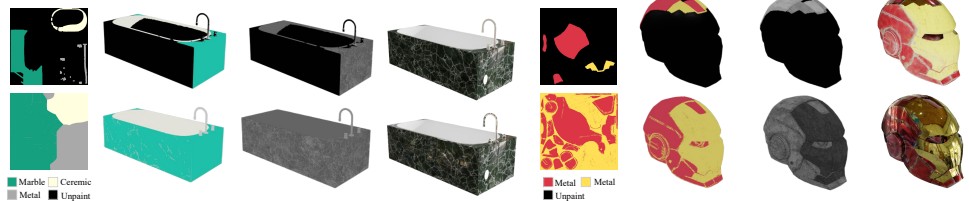

Figure 9: **Ablation study of missing part refinement.** Our method on the bottom row produces consistent texture maps and avoids the missing parts of material texture.

as plastic. Furthermore, the texture and appearance of materials also vary, such as the subtle wrinkles on the red box in the second column and the more pronounced color contrast at the base of the blue box, which enhances realism and reflects signs of use. Due to the albedo-referenced algorithm design in Section 3.2.3, regions with similar colors have similar BRDF values, avoiding abrupt changes in material properties, such as the continuous gold surface of the globe and the silver body of the kettle. Regions with significant color differences display consistent differential distributions with the key map, such as the embossed textures on the lower left corner of the water bottle in Figure 5 and the subtle particle variations on the surface of the red oil drum. Additionally, visualization of texture maps demonstrates reasonably consistent textures with albedo, shown in Figure 6. Quantitive results are shown in Table 1, and more qualitative results can be found in Appendix E.

**Texture refinement for generated 3D objects.** Figure 7 displays the qualitative results. By leveraging our model for enhancement, we observe that Make-it-Real successfully generates appropriate material maps for both image-to-3D and text-to-3D models.

Similarly, our Make-it-Real successfully paint these models with materials. As reported in Table 1, human experts consistently favor the objects post-refinement across all evaluated 3D content creation methods. This preference aligns with the evaluations performed by GPT-4V, indicating a general consensus on the enhancement in quality achieved through Make-it-Real refinement process.

### 4.3 Ablation Study

**Effects of mask refinement module.** In Section 3.2.1, we performed additional material post-processing on the segmentation outcomes, as depicted in Figure 8. The refined results in the last row indicate that our module achieves precise material segmentation for most standard objects, thereby enabling more accurate queries by GPT-4V. In Section 3.2.3, we addressed the completion of missing regions in the texture map within the UV space at the regional level, as illustrated in Figure 9. This method not only increases texture coverage but also enhances the visual quality and consistency of the refined 3D model. More ablation studies can be found in Appendix D.

**Effects of different texture maps.** We validate the impact of various texture maps generated by Make-it-Real on the appearance of 3D objects, as illustrated in Figure 16 of Appendix D.1. We provide a detailed qualitative analysis of how different maps enhance the visual texture of materials. For example, increasing metallic results in dampening of the base albedo and increasing in the shine on the surface, while reducing roughness gives the surface a smoother appearance and enhances highlights. Meanwhile, displacement and height maps contribute to the fine-grained bump details on the object's surface.

**Effects of different UV mappings.** Since UV mapping is a crucial step in the 2D-3D alignment technique of our method, we conduct experiments to assess how its quality impacts model performance. The results, shown in Appendix D.2 and in the second column of Figure 17, indicate that UV mappings with excessive fragmentation and color entanglement can cause issues. However, our method still performs well with artist-created UV mappings and Blender's built-in mapping techniques. This indicates our method still demonstrates good robustness with many mapping techniques.

## 5 Conclusion

In this paper, we present a novel framework leveraging MLLMs prior of the world to build a material library and proposing an automatic pipeline to refine and synthesize new PBR maps for initial 3D models, achieving highly photo-realistic PBR textures maps synthesis. Experimental results confirm that our approach can automatically refine both generated and CAD models to achieve photo-realism under dynamic lighting conditions. We believe Make-it-Real is a new and promising solution in the last few procedures of AI based 3D content creation pipeline with the development of MLLMs like GPT-4V [45] as well as the roaring field of deep learning based 3D generation from scratch.

## 6 Acknowledgement

This project is funded in part by Shanghai Artificial lntelligence Laboratory, the National Key R&D Program of China (2022ZD0160201), the Centre for Perceptual and Interactive Intelligence (CPII) Ltd under the Innovation and Technology Commission (ITC)'s InnoHK. Dahua Lin is a PI of CPII under the InnoHK.

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

# A    Appendix Overview

In this appendix, we provide additional details and results that are not included in the main paper due to the space limit. The attached video includes intuitive and interesting qualitative results of Make-it-Real.

# B    Details of Make-it-Real

In this section, we detail the pipeline that briefly outlined in the main paper. We commence by elaborating on the generation of SVBRDF Maps, incorporating illustrative figures to detail the process involving operations in computer graphics. Appendix B.1 is dedicated to explaining the acquisition of region-level texture map partition. Following that, in Appendix B.2 we discuss the method of pixel-level albedo-referenced estimation. Then, in Appendix B.3 we declare some details of rendering procedure. Appendix B.4 details the prompt design for material captioning and matching. Appendix B.5 reports the details of the GPT-4V based evaluation.

## B.1    Texture Partition Module Design

As illustrated in Figure 10, our process initiates with the rendering of a 3D object incorporating the original albedo (i.e. query albedo) from multiple perspectives. Following this rendering phase, we employ GPT-4V alongside a segmentor to derive segmented masks for the materials associated with each viewpoint. The subsequent step involves the extraction of regions masked in these images and their back-projection onto the mesh of the object. By examining the object from all acquired viewpoints and applying UV unwrapping techniques, we achieve preliminary segmentation of all materials. Subsequently, each material's segment is refined using a albedo-based mask refinement operation. Ultimately, by combining the segments of all materials, we obtain a region-level texture partition map, which serves to guide our subsequent, more detailed operations.

## B.2    Albedo-referenced Module Design

As illustrated in Figure 11, we developed a pixel-level albedo-referenced estimation module, building upon the foundations laid out in Appendix B.1. This module is inspired by a technique frequently employed by 3D artists, who often utilize albedo maps as a reference to generate images of other

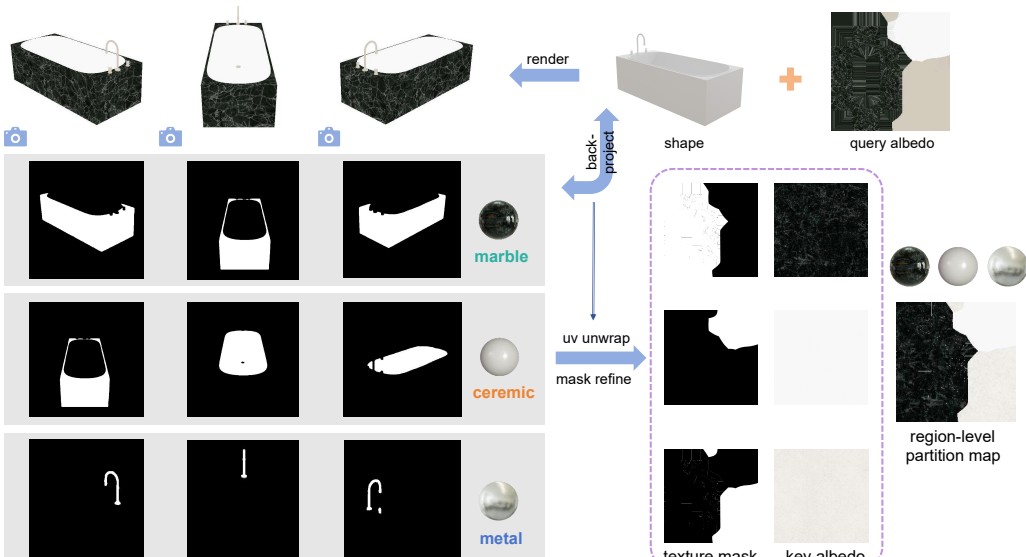

Figure 10: **Region-level texture partition module**. This module extracts and back-projects localized rendered images on to a 3D mesh, using UV unwrapping for texture segmentation, thereby resulting in precise partition map of different materials.

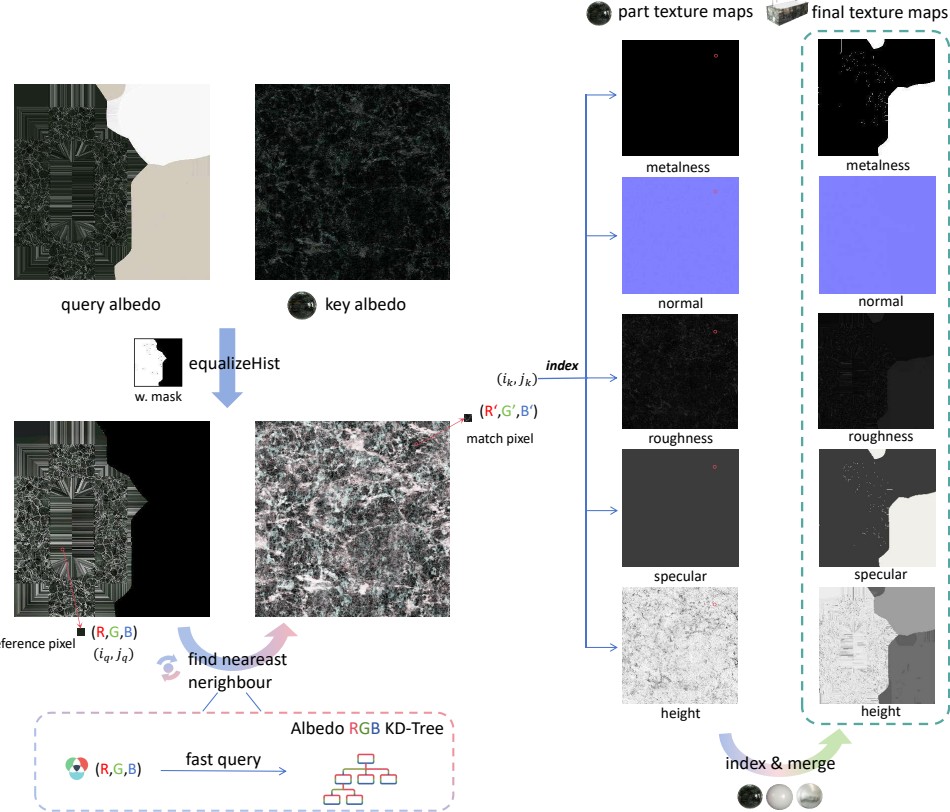

Figure 11: **Pixel-level albedo-referenced estimation module**. We generate spatially variant BRDF maps by referencing albedo maps, employing KD-Tree algorithm for efficient nearest neighbor searches, and normalizing colors via histogram equalization.

material properties. Accordingly, we designate the known albedo map as the query albedo, and the albedo corresponding to the region of interest in the material as the key albedo.

The process to precisely obtain the final SVBRDF maps is divided into four steps: 1) Initially, to address potential gaps in color intensity between the two albedo maps, histogram equalization is employed to achieve a more uniform color distribution across the image. 2) Subsequently, for each pixel on the query albedo—termed a reference pixel—we seek the most similar neighboring color index on the key albedo. Given the high dimensionality of both maps (typically 1024x1024 pixels), a brute-force approach to this search would be computationally prohibitive. To this end, we accelerate the pixel query process using a KD-Tree algorithm, which organizes the RGB values of the key map's albedo into a KD-Tree for efficient nearest neighbor searches, reducing the computation time to under ten seconds. 3) The third step involves using the obtained indices to obtain corresponding values from the rest of the material maps. 4) Finally, by aggregating the query results for all material segments, we are able to generate the comprehensive spatially variant SVBRDF maps.

## B.3 Rendering Procedure Details

In the context of computer graphics, "albedo" typically denotes the primary color of a material, a concept analogous to "base color" and "diffuse" in Physically-Based Rendering (PBR) paradigms, both representing the inherent color of the material under uniformly scattered illumination.

In accordance with workflow requirements, the inclusion of height maps, displacement maps, specular maps, and combinations of additional maps is optional. For objects demanding high surface details, such as the oil barrel in Figure 5 and the stone horse in Figure 16, displacement or height maps are incorporated and rendered using the mature 3D computer graphics software Blender engine. Regarding 2D-3D alignment techniques, including rasterization rendering, back-projection, and UV unwrapping detailed in Section 3.2.1, we adhere to the methodologies outlined in [51]. Specifically,

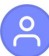
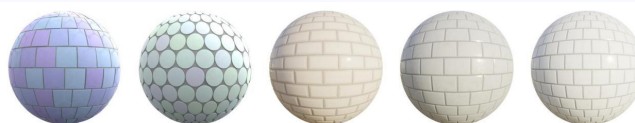

You are showed with a group of spherical PBR materials(all made of tiles, which is a kind of Ceramic, in total 5), can you generate a caption for each(about 20-30 words), try to distinguish their difference. Only describe the appearance features (must including color" and detailedmaterial'(such as patterns, roughness, metalness, concave and convex patterns, condition))and don't give too much other information. Do not describe whether it is reflective or not. Do notdescribe the shape of overall object. Please use a dictionary to represent the output result {1:Description 1, 2: Description 2, 3: Description 3, 4: Description 4, 5: Description 5)

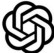

{1: "A gradient of pastel purple to blue tiles with a smooth, matte ceramic finish and slight bevel edges.",
2: "Soft teal ceramic tiles with a satin finish, arranged in a fish scale pattern with a slightly raised
texture.".
3: "Creamy beige ceramic tiles with a matte finish, laid in a staggered brick pattern with flatsurface and sharp edges.",
4: "Pure white ceramic tiles with a glossy finish, aligned in a straight grid pattern with a slightconvex curvature.",
5: "Bright white ceramic tiles with a matte finish, arranged in an offset pattern with pronounced grout lines and a smooth texture.}

Figure 12: **Material captioning**. We demonstrate how to construct detailed material descriptions, registering material information in textual form, thus providing a convenient bridge when querying with multi-modal large language models.

we utilize the Kaolin package (Kaolin [27]) and back-projection with masks [51], with the model learning rate set to 0.02 in Equation (3) and the difference coefficient $\sigma$ set to 0.1 in Equation (4).

This approach facilitates the systematic generation of various material texture maps in the presence of only an albedo map, thereby ensuring consistency and realism within the Physically Based Rendering (PBR) workflow. Moreover, the flexible application of different map types and 2D-3D alignment techniques significantly enhances the detail and realism of rendered objects, effectively meeting the demands of diverse rendering scenarios.

We evaluate the basic performance of the model by testing on a subset of 70 objects. Specifically, we measure the average runtime, the number of material parts per object, the number of GPT-4 queries per object, and GPU memory usage, as shown in Table 2. Additionally, we provide a detailed time consumption analysis for each step, presented in Table 3.

Table 2: **Model performance metrics.** The table reports the model's runtime, average number of material parts per object, queries per object, and GPU memory usage.

| Run Time (min) | # Material Parts/obj. | # Queries/obj. | Memory Cost (GB) |
|---|---|---|---|
| 1.54 | 2.42 | 5.76 | 12.52 |

Table 3: **Time usage statistics for each module.**

| | Render & Seg | Mat Retrieval | Mat Generation | Total |
|---|---|---|---|---|
| Run Time (min) | 0.26 | 0.49 | 0.79 | 1.54 |

## B.4   Prompt Details for Multi-modal Large Language Models

Prompts design has emerged as a pivotal factor for eliciting desired outcomes from MLLMs. This section delineates the intricacies of our prompt design for both material captioning and matching.

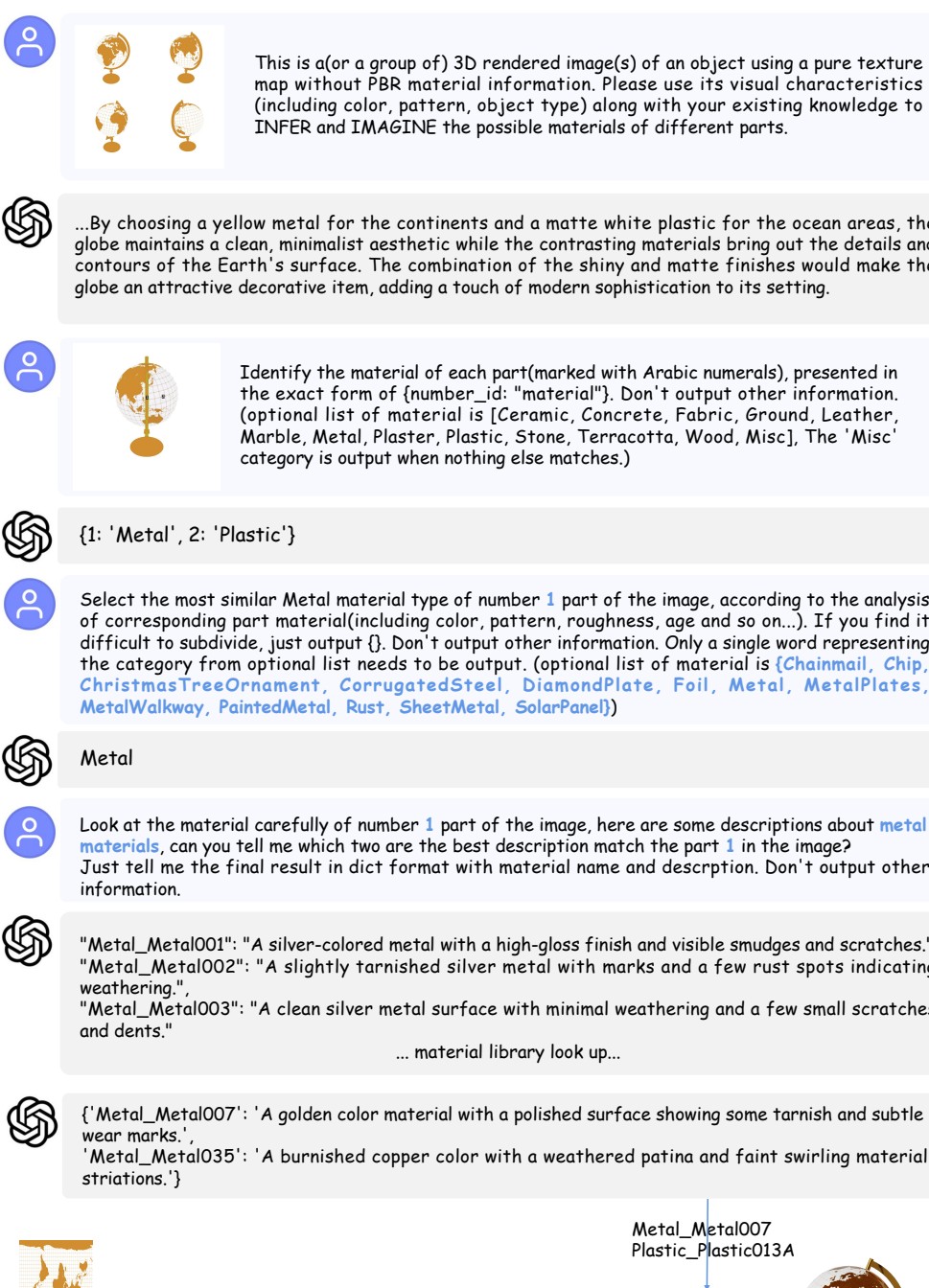

This is a(or a group of) 3D rendered image(s) of an object using a pure texture map without PBR material information. Please use its visual characteristics (including color, pattern, object type) along with your existing knowledge to INFER and IMAGINE the possible materials of different parts.

...By choosing a yellow metal for the continents and a matte white plastic for the ocean areas, the globe maintains a clean, minimalist aesthetic while the contrasting materials bring out the details and contours of the Earth's surface. The combination of the shiny and matte finishes would make the globe an attractive decorative item, adding a touch of modern sophistication to its setting.

Identify the material of each part(marked with Arabic numerals), presented in the exact form of {number_id: "material"}. Don't output other information. (optional list of material is [Ceramic, Concrete, Fabric, Ground, Leather, Marble, Metal, Plaster, Plastic, Stone, Terracotta, Wood, Misc], The 'Misc' category is output when nothing else matches.)

{1: 'Metal', 2: 'Plastic'}

Select the most similar Metal material type of number **1** part of the image, according to the analysis of corresponding part material(including color, pattern, roughness, age and so on...). If you find it difficult to subdivide, just output {}. Don't output other information. Only a single word representing the category from optional list needs to be output. (optional list of material is **{Chainmail, Chip, ChristmasTreeOrnament, CorrugatedSteel, DiamondPlate, Foil, Metal, MetalPlates, MetalWalkway, PaintedMetal, Rust, SheetMetal, SolarPanel}**)

Metal

Look at the material carefully of number **1** part of the image, here are some descriptions about **metal materials**, can you tell me which two are the best description match the part **1** in the image?
Just tell me the final result in dict format with material name and descrption. Don't output other information.

"Metal_Metal001": "A silver-colored metal with a high-gloss finish and visible smudges and scratches.",
"Metal_Metal002": "A slightly tarnished silver metal with marks and a few rust spots indicating weathering.",
"Metal_Metal003": "A clean silver metal surface with minimal weathering and a few small scratches and dents."

... material library look up...

{'Metal_Metal007': 'A golden color material with a polished surface showing some tarnish and subtle wear marks.',
'Metal_Metal035': 'A burnished copper color with a weathered patina and faint swirling material striations.'}

Metal_Metal007
Plastic_Plastic013A

mask projection    post-process    Material Generation

Figure 13: **Detailed prompts** of GPT-4V based material matching. Prompts in blue changes according to the current assigning part and GPT-4V's results.

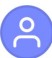
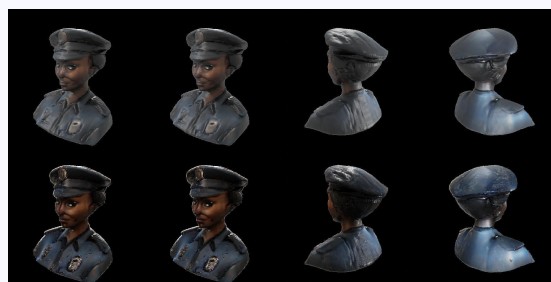

Assuming you are an expert in determining whether the textures of 3D models meet Photorealistic standards (mainly focus on the quality of texture), we have generated some 3D objects using two different methods. These objects have the same shape but different textures. Above are the four-view images of a model generated by the two methods (the upper half is the four views of the object generated by method one, and the lower half is the four views of the object generated by method two). Please help me evaluate which of the two objects is more Photorealistic in terms of texture. Respond with 1 if you think the upper half object is more realistic. Respond with 2 if you think the lower half object is more realistic. Also tell me the reason in a new paragraph.

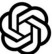

Based on the provided images, my evaluation would favor object 2 (the lower half object) as being more photorealistic in terms of texture.

The textures on object 2 have a more complex and nuanced appearance. The material of the uniform exhibits subtle variations in reflectivity and surface detail that suggest **a more realistic fabric texture**. Additionally, the **skin texture** appears more lifelike with variations in tone and reflectivity that are characteristic of human skin. The **metal police badges** on object 2 has a more nuanced reflection, which suggests a **more realistic interaction with environmental light**. In contrast, the textures on object 1, while still detailed, lack the same level of subtlety and variation. The uniform looks somewhat flat and less fabric-like, and the overall surface details are less convincing as natural variations.

Figure 14: **GPT-4V based evaluation prompts.** We define a prompt for GPT-4V to generate human-aligned comparison over objects before and after our texture refinement.

**Material captioning.** Due to limit context length for image tokens, it is currently impossible for MLLMs to directly memories thousands of images of material balls. To address this, we use GPT-4V to generate detailed captions for each material ball. As demonstrated in Figure 12, we input concatenated images of material ball from same subcategory with prompt specifically tailored to highlight texture properties. This strategy guides GPT-4V to generate detailed caption for each material ball, distinguishing the subtle differences between them. These detailed captions are then registered into material library.

**Material matching.** The MLLM-based material query process is exemplified through a simplified case, as illustrated in Figure 13. Initially, GPT-4V is queried to identify the basic material for matching. Following this preliminary matching, GPT-4V is queried for more specific materials using the names of subcategories as filters, which narrows down the selection to a few candidates within the same subcategory. For final material selection, GPT-4V is prompted with detailed captions from the library, directing it to allocate the most suitable material. This is accomplished in conjunction with a meticulously engineered segmentation process in UV space, ultimately facilitating MLLM-based material matching. It is noteworthy that prompting GPT-4V to navigate through a three-level tree structure has been proven to enhance both the efficiency and accuracy of the matching process, as opposed to directly selecting from thousands of materials directly.

## B.5 Evaluation Details of Quantitative Results

**Evaluation details.** For the purpose of evaluating objects pre- and post-refinement, we render four view images of each object and concatenate them vertically to facilitate a comprehensive assessment. We craft a specific prompt to guide GPT-4V in conducting an impartial comparison of the texture quality between two objects. As demonstrated in Figure 14. GPT-4V's advanced capabilities enable it to differentiate between the two objects, providing comparison results that closely align with assessments made by human experts. In the scenario evaluated, GPT-4V successfully identifies all enhancements (metal badges, human skin, and fabric textures).

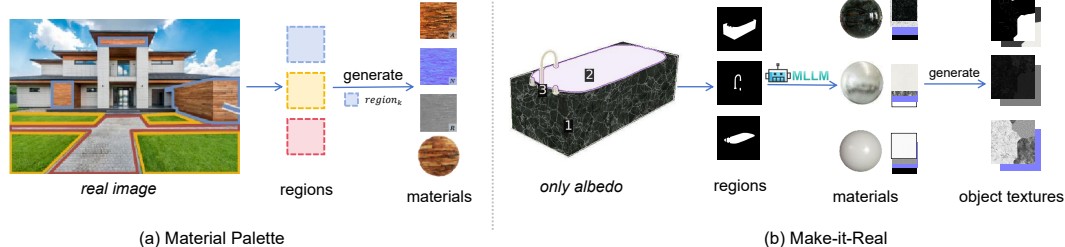

| (a) Material Palette | (b) Make-it-Real |
|---|---|

Figure 15: **Comparison between previous method and Make-it-Real.** We demonstrate the distinctions between Material Palette[40] and our method in terms of material identification and extraction. Our overall pipeline presents a more challenging task, where the input is a rendered image with only albedo information, and the output consists of textures for the entire object.

For human evaluators, we engage 16 volunteers with strong backgrounds in computer science, specifically in 3D modeling and generation. To ensure unbiased evaluations, we maintain gender balance through random selection. The assessment focuses on determining object material photorealism, as shown in Figure 14. Volunteers train on 40 Objaverse cases, with and without materials, to enhance their realism judgment for accurate test data evaluation.

## C   Additional Related Works

In the context of extracting real-world materials from single images, the methodologies most closely related to our work are Material Palette [40] and Photoshape [47]. Material Palette enables the extraction of materials at the region level from a single image, generating tilable texture maps for corresponding areas. Photoshape, on the other hand, automates the assignment of real materials to different parts of a 3D shape by training a material classification network. However, this approach needs the training of a material classifier, which is constrained to a limited set of object categories (e.g., chairs) and material types. Besides, both methods require rendered images of objects with real materials as input.

Our problem setting poses a more challenging task, as illustrated in Figure 15, which involves identifying and recovering the original material properties from images of objects that only have a albedo map, in addition to generating different material maps for the entire object. Humans have the capacity to intuitively infer the underlying material properties of 3D objects from single images containing only shape and basic colors. This capability is attributed to our robust material recognition abilities and comprehensive prior knowledge of object categories, colors, and material attributes. Building on this concept, our approach harnesses the potent image recognition capabilities and prior knowledge inherent in large-scale multimodal language models to efficiently execute region-level material extraction and identification, which is further enhanced by subsequent 2D-3D alignment and albedo-referenced techniques to generate and apply material texture maps for physically realistic rendering of 3D objects. Furthermore, [40] extracts the material takes 3 to 4 minutes, and process an image containing three materials takes more than 10 minutes. In contrast, our method takes only about 1 to 2 minutes to match and generate all the materials.

In more recent work, both MaPa [77] and our paper aim to enhance 3D objects with materials, but the key difference lies in material retrieval methods. MaPa [77] relies on CLIP, while we use MLLM, a distinction that we believe significantly boosts retrieval effectiveness for these reasons: **1)** Enhanced Global Context: CLIP focuses on one area at a time, requiring other parts to be masked, which limits global semantic information. MLLM, however, retains this context by simply highlighting the relevant area. **2)** Hierarchical Material Dictionary: We introduce a hierarchical dictionary in MLLM, providing descriptions and images of material spheres, enabling access to a more extensive library of 1,394 materials compared to MaPa's 118. **3)** Advanced Explainability: With MLLM's rapid development, transitioning from CLIP to a multi-step hierarchical inference process with LLM offers improved performance and clearer decision-making.

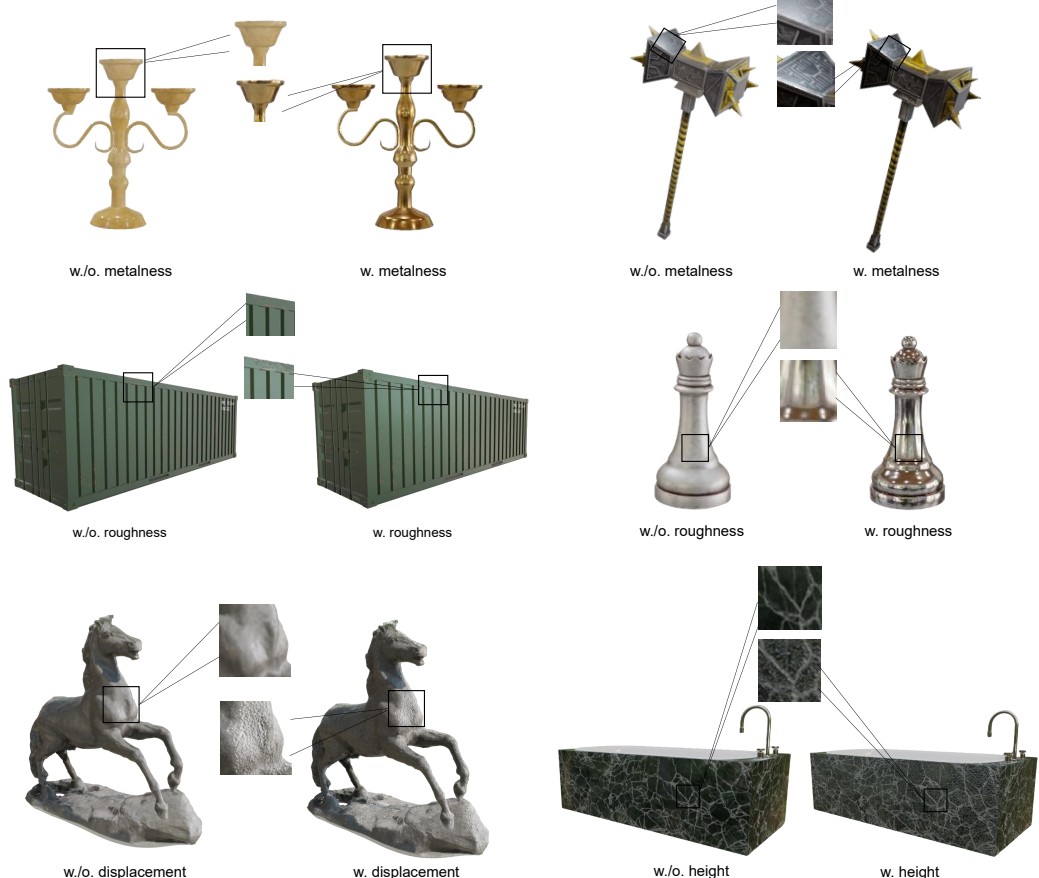

| w./o. metalness | w. metalness | w./o. metalness | w. metalness |
| w./o. roughness | w. roughness | w./o. roughness | w. roughness |
| w./o. displacement | w. displacement | w./o. height | w. height |

Figure 16: **Effects of different texture maps.** We evaluate the effects of metalness, roughness, and displacement/height maps on the appearance of 3D objects.

# D  Additional Experiments

## D.1  Effects of Different Texture Maps

We validate the impact of various texture maps generated by Make-it-Real on the appearance of 3D objects, as illustrated in Figure 16. For each example, the first column lacks the corresponding map enhanced by Make-it-Real, while the second column includes the same conditions with the specific material map. Initially, we compare the effect of metalness on object appearance. We observe that the objects on the right, with a higher metalness map value, exhibit higher reflectivity, and the reflected light's color is similar to the material's own color, closely resembling real-world objects and appearing more aesthetically pleasing. In contrast, the objects on the left without the metalness map have lower reflectivity, and the reflected light tends to be white.

Next, we examine the impact of the roughness map, which controls the smoothness of the material's surface. We observe that the silver chess piece on the right, with a low roughness texture map, becomes smooth, and together with metalness, produces a mirror-like reflection effect. On the other hand, the box on the left with a roughness map exhibits changes in light dispersion on its surface, with some areas showing highlights, while also adding and enriching scratch texture details on the surface.

Furthermore, we compare the effects of displacement and height on objects, both of which are usually optional and can also impact object appearance. Height maps typically use grayscale values to represent the surface's relative height, simulating a relief effect through changes in lighting and shadows. As shown in the last row on the right, the object with a height map has an uneven surface, enhancing the sense of depth. Displacement mapping is a more powerful technique that changes the

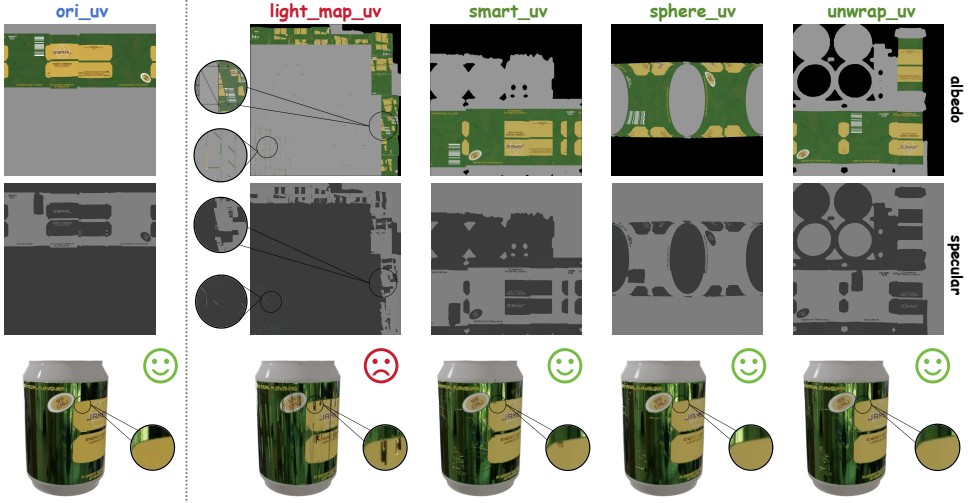

Figure 17: **Effects of different UV mappings of input mesh.**

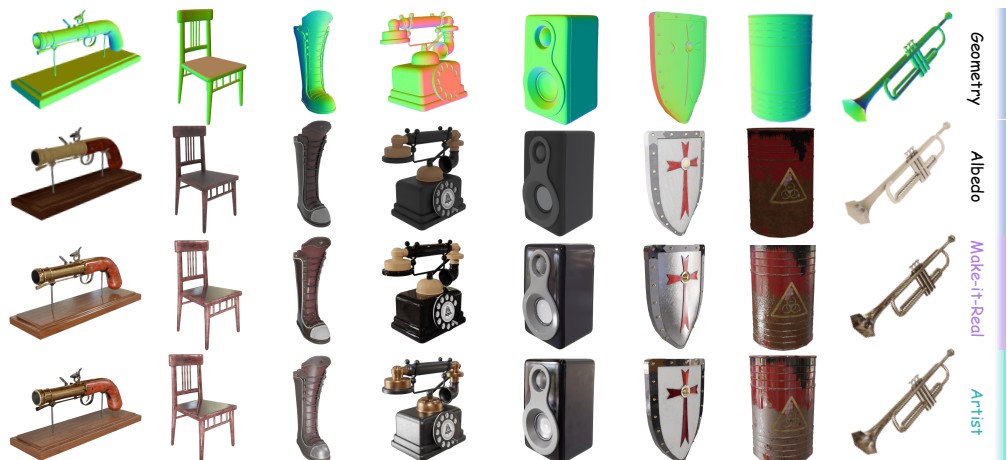

Figure 18: **More Comparisons between Make-it-Real and Artist-Created Materials.**

vertex positions of the geometry based on the map's values, creating realistic relief effects. As shown on the left, the stone sculpture with displacement mapping exhibits very realistic details and a sense of relief. We acknowledge that the generated PBR maps are not true representations of physical maps, but our algorithm generates PBR maps that closely approximate the visual characteristics of true PBR properties, which significantly enhances the realism of 3D objects.

## D.2 Effects of Different UV Mappings

We conduct experiments to evaluate the impact of UV mapping on our method. As shown in the second column of Figure 17, UV mappings with excessive fragmentation and color entanglement can cause the 2D segmented images to mix with other regions when reprojected into the UV space, leading to material blending issues. However, our method shows good results with original artist-created UV mappings and Blender's built-in mapping methods in Figure 17, such as *smart*, *sphere*, and *unwrap*. This indicates that our method still demonstrates good robustness with many mapping techniques.

In practice, we observe that most objects in Objaverse have UV mappings with good properties, meaning they are not excessively fragmented into small pieces. Additionally, we can control the UV mapping process: For some low-quality UV maps and generative objects that originally lack UV maps, we can re-unwrap them to achieve higher quality.

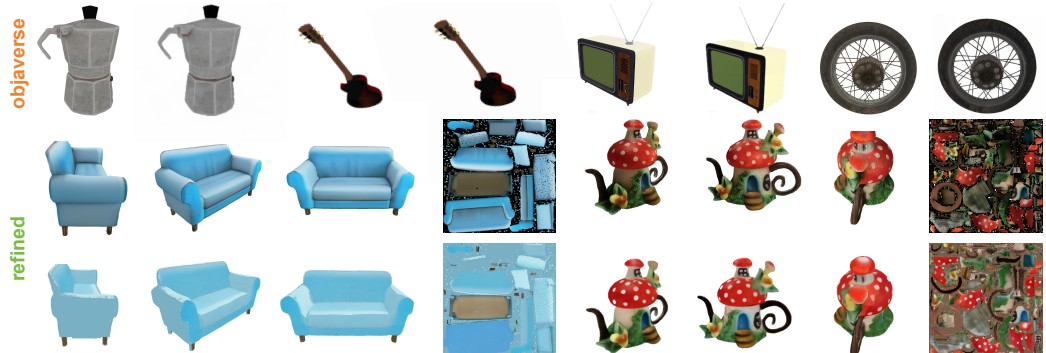

Figure 19: **Addressing Shading Issues with IntrinsicAnything [9].** The first row shows that the derendered albedo images from Objaverse are consistent with the original one. The last two rows demonstrate the successful derendering results for generative models, effectively reducing shading.

# E   More Qualitive Results

## E.1   Make-it-Real for Existing 3D Assets

In this section, we show more qualitative results of Make-it-Real for existing 3D assets from [16]. We show more qualitative results of Make-it-Real for existing 3D assets from [16]. Results are shown in Figures 20 and 21. The first row presents the original 3D object with only a albedo map, while the second row showcases the object enhanced by our method with various material maps.

**Comparisions with Artist-Created Materials.** We conduct a user study comparing materials generated by our Make-it-Real with those created by artists, visualizing some results in Figure 18 for close examination. For objects like pistol, wooden chair, speakers, and saxophone, Make-it-Real showed strong similarity to artist-made materials, maintaining consistent metallicity, roughness, highlights, and coloration. In cases such as shield, boots, landline phone, and oil drum, our method produced materials that, while different, are realistic and sometimes visually superior. Our pipeline approaches the level of refinement seen in some artist-created materials, which often require significant time and effort, and clearly outperforms more basic and crude artist-generated materials. This highlights its effectiveness and potential for practical applications.

## E.2   Visualization of generated texture maps.

In this section, we show visualization of generated texture maps in Figures 22 and 23. Our approach, guided by the query albedo reference, produces material maps with distinct material partitions and maintains a distribution consistent with the albedo map. As a result, the enhanced object exhibits both realism and texture consistency.

# F   Limitations and future work

While promising results have been achieved by MLLMs for texture assignment through Make-it-Real. Our work still faces several challenges.

**Lighting Effect and Addressed Solutions.** First, although our method can achieve appropriate texture assignment for albedo-only model, it does not support reverse transform from shaded texture map to albedo map for generated 3D objects. This causes problem of assigning different materials to the dark shadow and highlight area when generated object with mesh already shaded in different lighting conditions.

This issue is less pronounced in artist-created models, where albedo maps have less shadows and lighting effects. Our tests show that the albedo obtained from inverse rendering methods [9] closely matches the original artist-created albedo maps from the Objaverse dataset, as illustrated in the first row of Figure 19. For generative models, baked-in shading effects are more pronounced. Our method addresses this by integrating mature derendering algorithms [9, 14, 31, 66], specifically the Intrinsic Anything [9], into our pipeline with minimal complexity. This integration derenders from four

viewpoints and back-projects to obtain a better albedo map, then applies our Make-it-Real method for material painting. As shown in the last row, this approach reduces lighting noise and supports a wider range of inputs.

**Impact of Model Quality and Segmentation.** Second, we find base model quality is essential for MLLM to assign correct materials. When base 3D object is of low quality(e.g., uneven surfaces or mixed colors across different parts), it is difficult for MLLMs to identify object properties when ground truth text prompt describing the object is not available. Additionally, there has been limited progress in material segmentation in 2D and 3D demoain [52, 38]. The material segmentation algorithm included in our proposed pipeline can serve as a useful baseline and provide inspiration for future work. We also explore the integration of the pipeline with 3D segmentation networks, which shows promising results. For future work, we consider methods mitigating these challenges as well as adding user friendly control into our fully-automatic pipeline.

# G    Data Ethics

The datasets utilized in our work, specifically Objaverse, is composed entirely of inanimate objects. Importantly, the Objaverse dataset we used have undergone rigorous ethical filtering as part of the Cap3D [41]. Cap3D's ethical filtering process removed all potentially problematic data, including any identifiable human elements and NSFW content (detailed in Sec 3.2 of Cap3D). This process included the removal of objects with licenses that do not permit commercial usage, the exclusion of objects lacking sufficient camera information for rendering, and the application of face detection and NSFW classifiers with high thresholds to ensure thorough filtering. The final dataset used in our work, therefore, does not contain any human-derived data or data related to human subjects.

We recognize the potential risk of misuse associated with this technology, particularly in the creation of realistic fake human representations. However, since our dataset fully excludes human data, this risk is substantially mitigated.

# H    Broader Impacts

**Potential positive societal impacts.** The proposed method facilitates more realistic and accurate representations of materials in 3D models, benefiting industries such as gaming, virtual reality, and film, leading to more immersive and engaging experiences. By automating the material assignment process, Make-it-Real significantly reduces the time and effort required for 3D content creators, allowing for more efficient workflows and enabling creators to focus on more creative aspects of their work. This approach can democratize high-quality 3D content creation by making advanced material application techniques accessible to a broader range of users, including those without specialized skills in graphic software.

**Potential negative societal impacts.** The improved realism in 3D assets could be exploited for creating highly convincing fake visuals or deepfakes, which might be used in disinformation campaigns or to mislead audiences. There is a risk that the materials generated could inadvertently perpetuate biases or stereotypes if the training data for GPT-4V includes biased representations of certain materials or objects. As the method involves processing and recognizing visual data, there could be concerns regarding the privacy of any real-world images used as inputs, particularly if they contain sensitive or personal information.

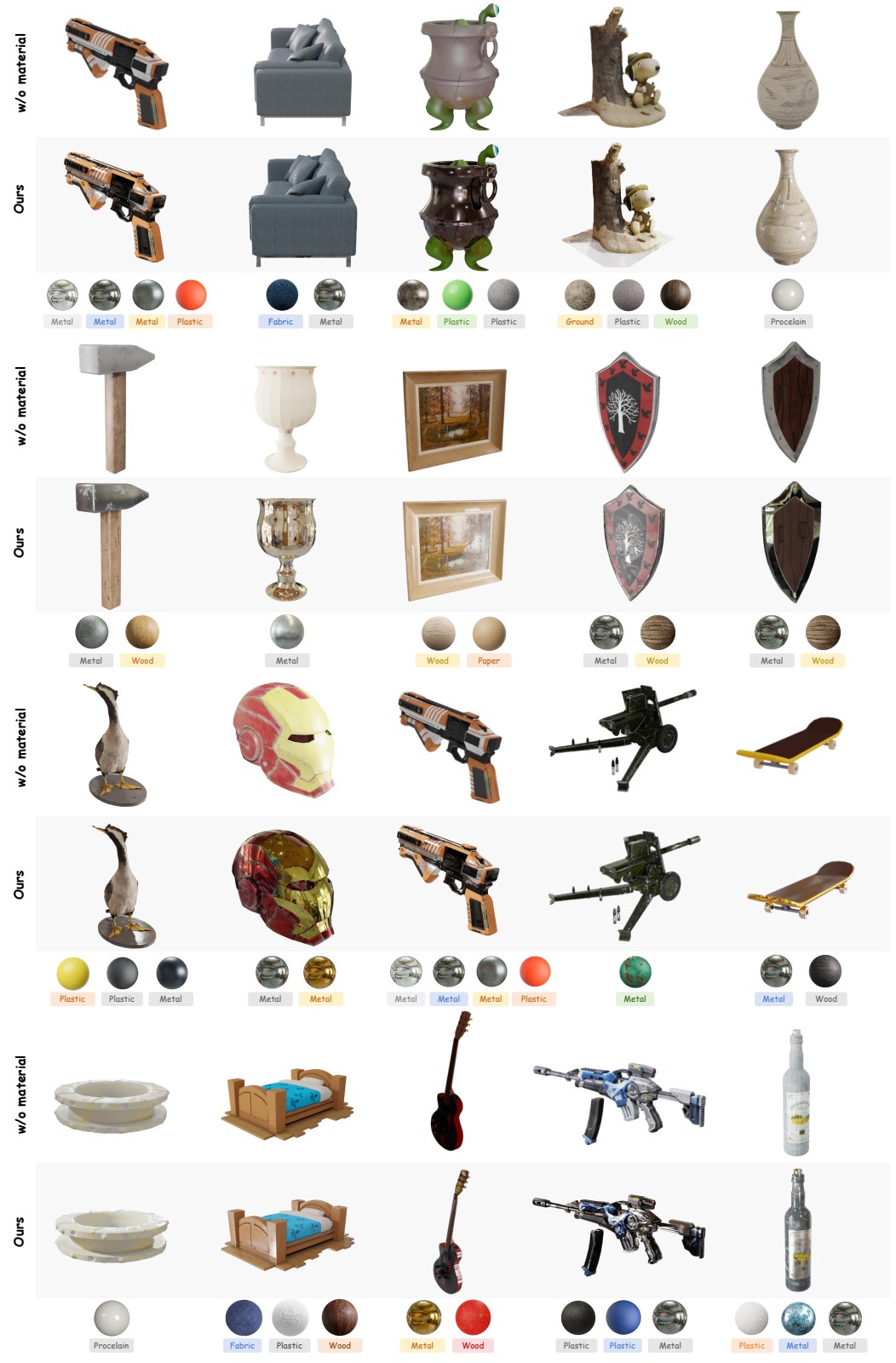

Figure 20: **More qualitative results of Make-it-Real refining existing 3D assets without material.** Objects are selected from Objaverse[16] with albedo only.

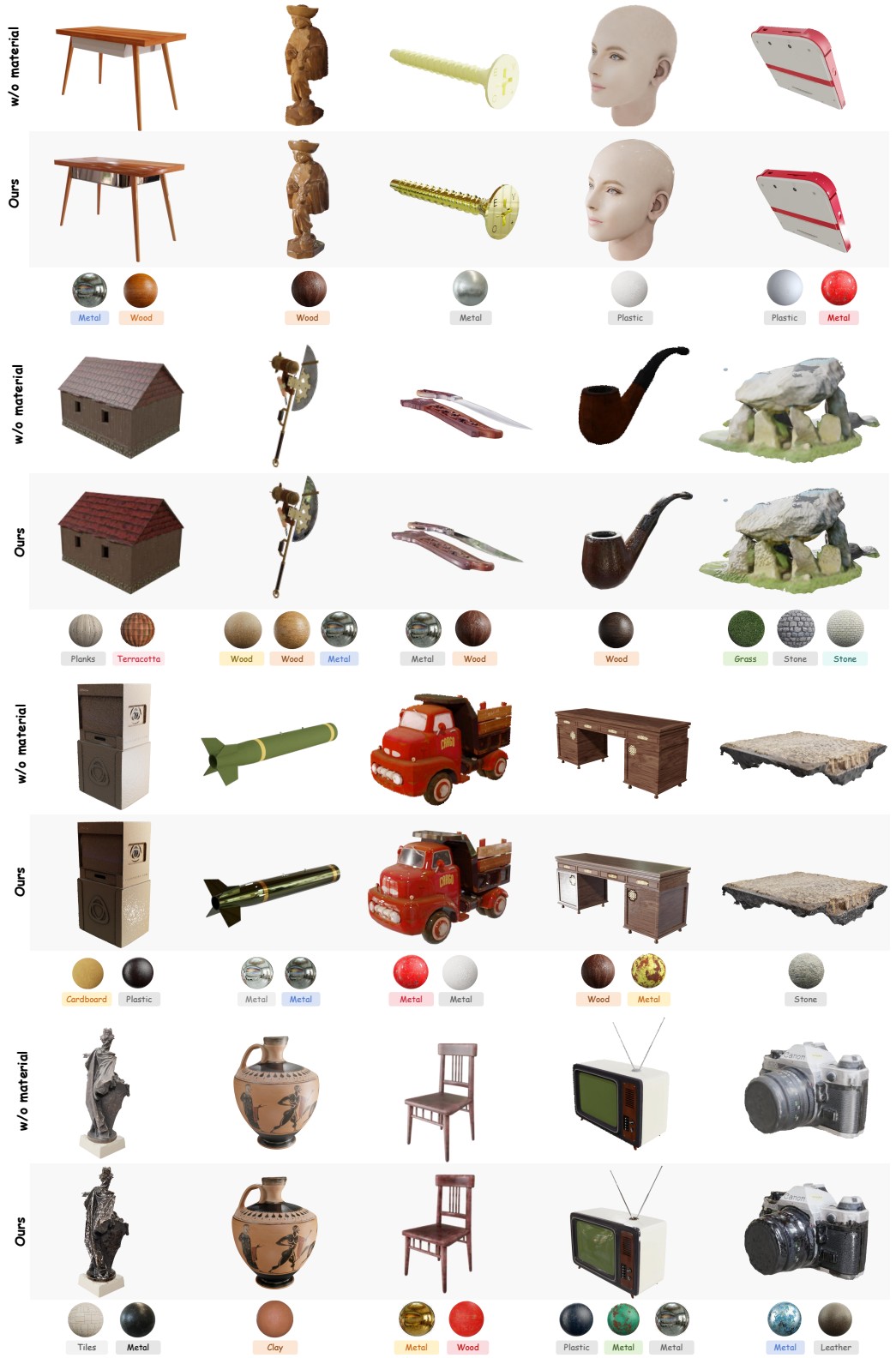

Figure 21: **More qualitative results of Make-it-Real refining existing 3D assets without material.** Objects are selected from Objaverse[16] with albedo only.

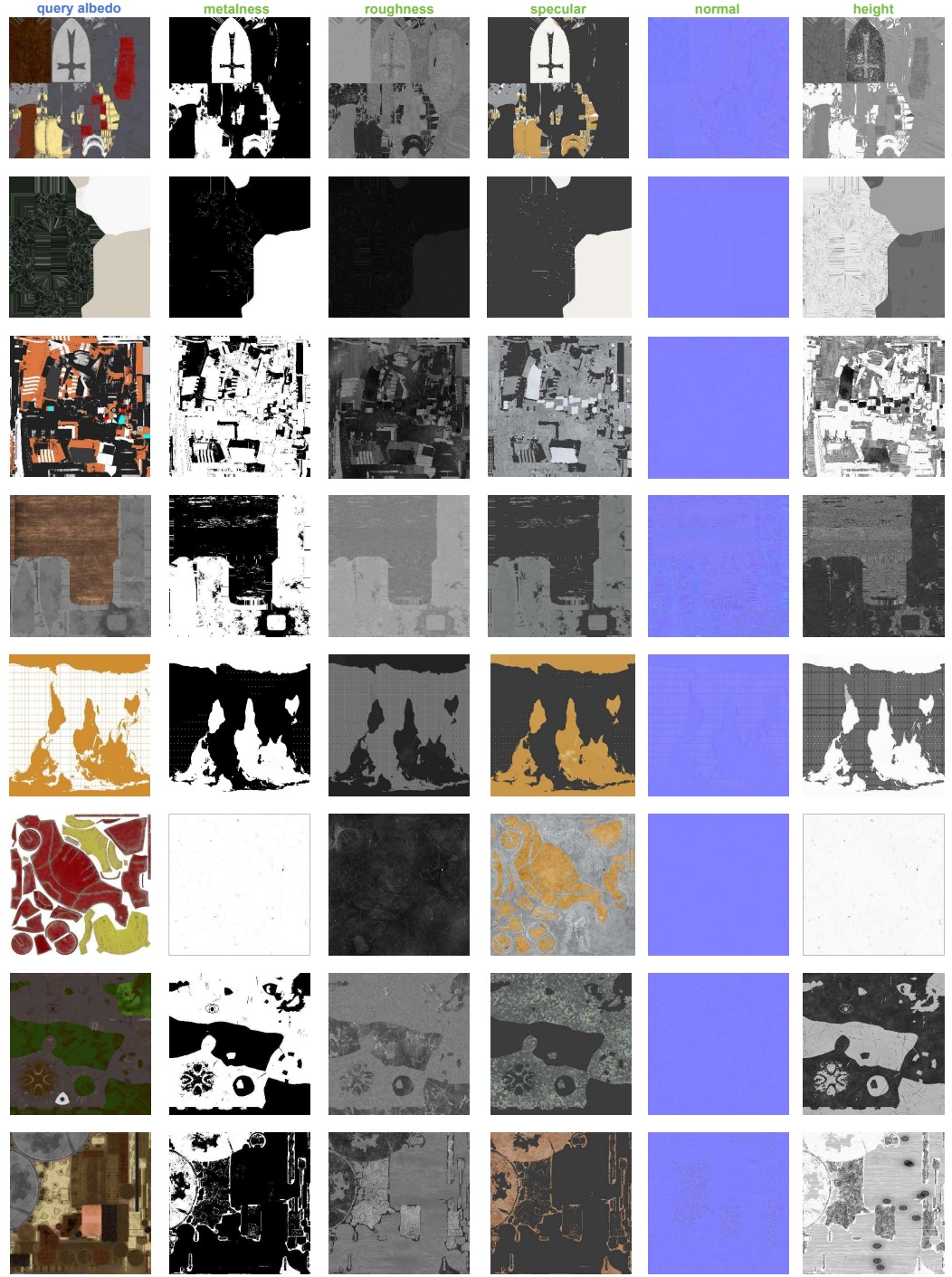

Figure 22: **Visualization of generated texture maps.** The first column represents the original query albedo map of 3D objects, while the subsequent columns showcase the corresponding material maps generated by Make-it-Real.

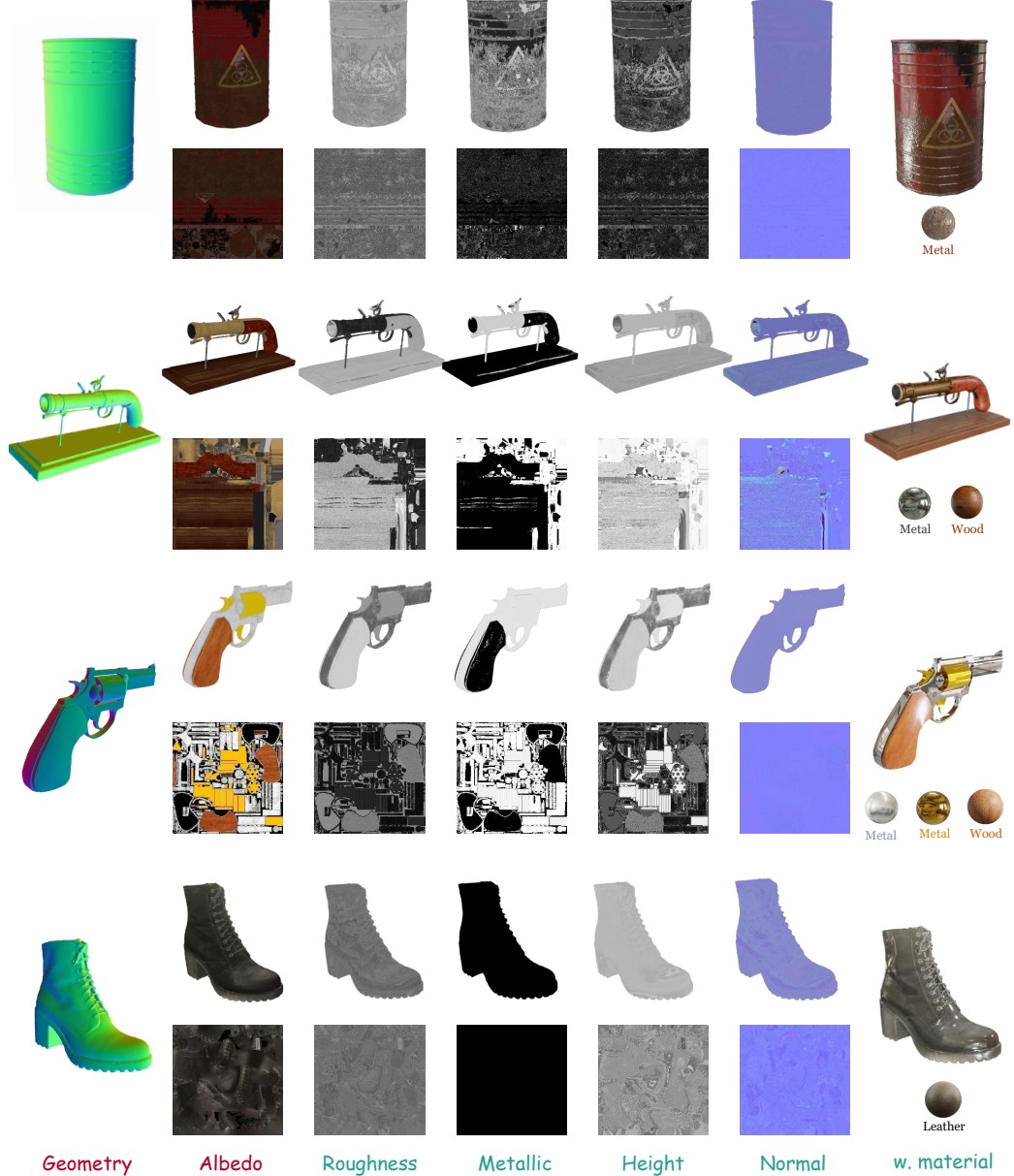

Geometry    Albedo    Roughness    Metallic    Height    Normal    w. material

Figure 23: **Visualization of generated texture maps.** We show part of the SVBRDF material maps generated by Make-it-Real and the final rendering results. We displayed the texture maps and the corresponding 3D rendering effects. The albedo is the input, and the following four columns show the material effects in the UV space and on the 3D object.

