# OpenReview forum: "Make-it-Real: Unleashing Large Multimodal Model for Painting 3D Objects with Realistic Materials"
_NeurIPS.cc/2024/Conference — NeurIPS 2024 poster_

### Official Review · Reviewer_jx7i · 2024-07-07

**Soundness:** 2
**Presentation:** 1
**Contribution:** 1
**Rating:** 3
**Confidence:** 5

**Summary:**

This paper aims to assign material information to 3D objects that already have a base color. They do not generate material maps through training generative models but instead use a pre-established material library and build an automated recognition and indexing system to index from the material library. To use this framework, the authors pre-built a material library and proposed some prompt templates to prompt GPT-4V. In this process, the segmentation of parts is mainly performed using semantic-SAM. The authors iteratively segment different views and eventually fuse them into a single UV map.

**Strengths:**

1. The core of the entire system is retrieval rather than generation from a generative model. This approach generally offers better controllability, adjustability, and interpretability.

2. Some techniques to accelerate querying are proposed, such as hierarchical prompting.

**Weaknesses:**

1. This task is very limited because it requires a 3D mesh with an already existing base color. The requirement for a base color is not very practical. In line 113, the authors mention, "Given the advancements in generating high-quality 3D shapes with albedo maps, the restoration of realistic material properties remains a challenge." I disagree with this point. One of the main difficulties in current 3D generation or texture generation methods is generating a base color (lightning, shadow-free). Most methods, whether through 2D SDS or 3D rendering training, fail to generate the base color. The more pressing issue is actually estimating albedo and material properties from the final RGB image.

2. In line 136, the authors mention, "These patches are then merged based on similar colors to obtain the final material grouping." This is very limiting, as it implies that parts with the same base color are assumed to have the same material.

3. The technical contributions of the paper are minimal, mainly leveraging the capabilities of GPT-4V for VQA. I do not believe designing prompt templates is a sufficient contribution. Additionally, it is hard to reproduce the authors' implementation since GPT-4V is close-sourced. I believe the authors should at least demonstrate the feasibility with another open-source VLM model.

4. Most of the key implementation details are placed in the supplementary file. However, the step of retrieving the corresponding material based on the given albedo map and the material library established by the authors is crucial (i.e., section B.2). At the very least, this part needs to be included in the main text. Additionally, the procedure is hard to understand (please see my questions in the "Questions" section).

5. I have some concerns about the evaluation of this task because it is difficult to assess the ability to assign materials. From the results provided by the authors, it is hard to see a significant difference between different parts. It seems that compared to the albedo-only baseline, the main difference is the more pronounced lighting effect caused by the higher metallicity in the authors' results. Besides, I feel the quality is poor from the video demo at 00:56.

6. This work is more related to texture and material generation, rather than performing inverse rendering from existing photo-realistic rendered images. In the related work section, studies on texture generation need to be mentioned.

**Questions:**

1. In line 39, the authors mention "Additionally, shadows and lighting can affect judgment." I cannot understand how shadows and lighting come into play here. If the authors assume that only the base color is used, the view can be directly obtained through rasterization without the influence of shadows and lighting.

2. For the section B.2, what is the key diffuse? Is it an existing diffuse map from your library? If this is the case, how to select the key diffuse based on the current query diffuse? Besides, what is "3) The third step involves using the obtained indices to obtain the corresponding
533 values from the rest of the material maps."? Please elaborate more about this part.

**Limitations:**

The authors point out the limitations of their work in the supplementary material and leave it for future work.

---

> ### Author Rebuttal · Authors · 2024-08-06
>
> Thank you for your detailed review and insightful comments. We will address your questions and concerns below and in the revised paper.
>
> ***
>
> ### W1. Importance of Addressed Task for Material Painting
>
> While the derendering task mentioned by the reviewer is important, we believe the task of adding materials to albedo-only 3D objects is also essential for reducing the extensive time 3D artists spend creating realistic texture maps. Although graphics tools like ShaderMap integrate this functionality, they are limited to producing simple lighting effects. Our method fully explores the potential of MLLMs to accurately capture complex materials and provide a solution for realistic, part-level material generation.
>
> Additionally, we believe that derendering and material restoration are not mutually exclusive fields but rather complement each other. Research in the latter area can serve as a tool for automating the generation of 3D material data and help researchers gain a deeper understanding of material properties and their realistic effects in the physical world.
>
> ***
>
> ### W2. Discussion of Material Grouping Method
>
> Thank you for pointing out. We acknowledge that, in theory, the assumption that parts with the same base color have the same material can be limiting. However, in our experiments, we found that in most cases in the dataset, semantic patches obtained through SemanticSAM tend to have similar colors when they represent the same material. Based on this observation, we implemented a merging process, carefully controlling the merging threshold based on experimental results. This approach reduces the number of GPT-4V queries without compromising effectiveness. We will include more details about this in the final version.
>
> ***
>
> ### W3. Unleashing GPT-4V's Ability for Material Painting Beyond VQA
>
> Our core contribution is not in applying multimodal methods for VQA but in effectively leveraging and fully exploring the potential of MLLM to achieve an innovative and practical application: material painting. GPT-4V, even with well-designed VQA tricks, cannot perform material painting on objects because it is confined to the 2D space. By designing a comprehensive pipeline, we aim to unlock MLLM's ability to understand the 3D physical world and generate realistic material representations.
>
> Our pipeline include an advanced segmentation method in the UV space of materials, a rich set of visual and text cues for faster material retrieval, and an innovative algorithm that generates SVBRDF material maps using a region-to-pixel approach. This integration bridges 2D MLLM models and the 3D physical world, empowering large language models with the capability to paint materials to objects at the part level.
>
> ***
>
> ### W4. Key Implementation Details
>
> You are correct that Appendix B.2 contains the key implementation details of our method. In Section 3.2.3, we describe the essential aspects of SVBRDF map generation in text form and refer to figures in the appendix to aid understanding. However, we acknowledge that the main text contains limited illustrations for this section. We will enhance the discussion and include more illustrative figures in the final version to improve clarity.
>
> ***
>
> ### W5. Material Effect of Presented Results
>
> Materials are inherently complex physical properties, and differences can be seen by zooming in on different regions, such as the globe, bathtub, and shield in Figure 5. I would like to clarify that the material effect is not solely due to higher metallicity producing a brighter lighting effect. Instead, it results from the interaction of multiple PBR maps: increased metallicity results in dampening of the base albedo and increased surface shine, increased roughness reduces specular highlights, and displacement and height control fine surface details. These values are assigned based on the albedo, type, and local region of the object, so not all parts have the same higher metallic settings. (More infomation in Figure 16)
>
> Regarding the video demo, we're sorry for the quality of the rendered scene at the end frame. Due to significant memory requirements for rendering scenes, this picture may not fully reflect the clarity and precision of the objects. We recommend referring to the clearer examples in the main part of the video and paper.
>
> ***
>
> ### W6. The Clarification on Related Works
>
> We have cited and discussed several texture generation works in the *Material Capture and Generation* subsection in Sec 2. Related Work. We introduce recent works on texture generation, such as Paint-it and Collaborative, which generate various material texture maps based on shape. We also reference earlier works like Fantasia3D and Matlaber. We discuss the approaches and limitations of these methods, and we will update the final version with more related works in this area.
>
> ***
>
> ### Q1. The significance of identifying shadows and lighting
>
> Sorry for any confusion. We would like to clarify that besides retrieving materials for albedo-only 3D objects, our approach involves describing rendered thumbnails of real materials in the Material library to construct the dataset annotations. These thumbnails may include lighting and shadows. The task in our whole setting requires two capabilities of recognizing materials: 1. Accurately describing real materials under ambient lighting, and 2. Inferring local materials from distorted albedo. We will add more explanations about this point in the final version.
>
> ***
>
> ### Q2. Method of Key Diffuse and Indexing
>
> The key diffuse refers to the diffuse map of the material retrieved by the MLLM. This can be understood through the multiple arrows stemming from `index` in Figure 11. After obtaining the nearest neighbor coordinates in step 2), we use the coordinates to index the values of the same position in other material maps paired with the current diffuse. We will draw more illustration(like serial number) on our figure to make it more clear.
>
> ***

---

> > ### Comment · Reviewer_jx7i · 2024-08-11
> >
> > Thank authors for the rebuttal. Although some technical details were clarified, my main concern remains unresolved and still exists. That is, the assumptions about the material are overly simplistic, and most of the contributions come from the use of the GPT-4v, with not much application of knowledge related to materials. Besides, the visual quality is not good enough, especially in the case that an existing diffuse map is assumed to be provided.
> >
> > Currently, there has been an abundance of work that leverages multimodal models to assist in addressing specific domain issues. However, this paper’s design in the area of domain knowledge is weak (i.e., the assumptions are too simplistic). This leads me to believe that the technical contributions of this paper are insufficient for acceptance.
> >
> > I respect the opinions of the other reviewers, but I still maintain my opinion.

---

> > > ### Author Response · Authors · 2024-08-12
> > > **Thanks for the response**
> > >
> > > Thank you for your continued engagement with our paper. We appreciate your comments and would like to respond to the concerns you have raised.
> > >
> > > We understand your concern regarding the assumption of an existing diffuse map. Diffuse-only objects are widely found in datasets like Objaverse-XL[1], Objaverse[2], and Shapenet[3] because creating them is relatively easy. However, generating additional material maps from a diffuse map remains a time-consuming step. Our approach significantly accelerates the materially realistic 3D content creation process. Moreover, our method can be easily complemented with existing derendering techniques when shadows exist in the diffuse maps (e.g., generated objects), as shown in Figure C3 of our rebuttal PDF. These techniques are orthogonal and can work well together to meet different requirements and scenarios, as mentioned in our response to W1.
> > >
> > > Concerning the contribution besides utilizing GPT-4v, we acknowledge the significant potential and prior knowledge that GPT-4v brings. However, it alone is insufficient to accomplish our goals. We develop an entire pipeline, comprising texture segmentation, material matching, and SVBRDF map generation, that introduce domain knowledge and abilities far beyond GPT-4v.
> > >
> > > Regarding visual quality, we recognize that there is room for improvement. In the meantime, few existing methods can automatically generate comprehensive material maps, particularly for displacement and height maps. We wish our Make-it-Real can offer new insights to accelerate artistic creation.
> > >
> > > Thank you once again for your time and valuable feedback. Your insights provide us with important considerations for improving our work, and we look forward to incorporating these enhancements as we continue our research.
> > >
> > > ---
> > >
> > > [1] Deitke M, Liu R, Wallingford M, et al. Objaverse-xl: A universe of 10m+ 3d objects.
> > >
> > > [2] Deitke M, Schwenk D, Salvador J, et al. Objaverse: A universe of annotated 3d objects.
> > >
> > > [3] Chang A, Funkhouser T, Guibas L, et al. Shapenet: An information-rich 3d model repository.

---

### Official Review · Reviewer_dDLL · 2024-07-09

**Soundness:** 3
**Presentation:** 3
**Contribution:** 3
**Rating:** 5
**Confidence:** 3

**Summary:**

This paper presents a novel framework leveraging MLLM priors (GPT-4V) to build a material library and proposes an automatic pipeline to refine and synthesize new PBR maps for initial 3D models with diffuse albedo only. The pipeline integrates existing tools, such as GPT-4V and Semantic-SAM, while introducing novel techniques to refine and complete segmented masks, resulting in a set of full SVBRDF maps. Experimental results demonstrate that this approach can automatically refine both generated and CAD models to achieve photorealism under dynamic lighting conditions.

**Strengths:**

1. The provided results exhibit a decent quality, particularly for objects with metallic or specular materials
2. The paper is well-written, covering all necessary details for reproduction.
3. The potential application is clear: it can be used in a plug-and-play manner for 3D generative models to enhance their photorealism.

**Weaknesses:**

1. The novelty is kind of limited. The major components rely on powerful backbones such as GPT-4V and Semantic-SAM, with the novel contributions primarily found in the segmentation and material retrieval parts.
2. The (quantitative) evaluation is sort of limited, with only GPT-4V-based and user-based study results reported in Table 1. Given that this is a novel application without standard metrics, I will not critique it too harshly.
3. The quality of the results depends heavily on the initial albedo map. If the given albedo map is not clean (e.g., it contains baked-in lighting effects), the proposed method cannot correct it, resulting in weird SVBRDF maps.

**Questions:**

Generally speaking, I have no further questions regarding this submission; my concerns are already listed above. While the novelty might be somewhat limited, the clear potential applications and its widespread applicability lead me to still stand on the positive side of this submission.

**Limitations:**

Limitations and potential negative social impact are well discussed.

---

> ### Author Rebuttal · Authors · 2024-08-06
>
> Thank you for your detailed review and insightful comments. We will address your questions and concerns below and in the revised paper.
>
> ***
>
> ### W1. A clarification of novelty of Make-it-Real
>
> Thank you for the comments. Our key innovation lies not in the MLLM model technology itself, but in how we effectively leverage and fully explore the potential of MLLM to achieve innovative and practical applications. MLLM alone cannot perform material painting on objects, so we develop a practical and comprehensive pipeline to unlock this capability.
>
> Our core contributions include an advanced segmentation method in the UV space of materials, a rich set of visual and text cues for faster material retrieval, and an innovative algorithm that generates SVBRDF material maps using a region-to-pixel approach. This integration bridges 2D MLLM models and the 3D physical world, empowering large language models with the capability to accurately assign materials to objects. The generated materials are compatible with various rendering engines, enhancing the versatility and utility of our approach in creating realistic 3D assets.
>
> ***
>
> ### W2. About quantitative evaluation of painting materials
>
> Thank you for your understanding regarding this novel application. Previous work primarily focused on 3D object generation without explicitly separating base color from different PBR properties. However, there is now a growing interest in enhancing the physical material properties of 3D objects, yet evaluation methods remain limited.
>
> Obtaining ground truth for material properties is challenging mainly due to the ambiguity in material ground truth for certain objects; for example, a white albedo mug could be made of metal or ceramic, leading to different material maps for the same albedo input. Given this ambiguity, we use the evaluation methods described in GPT4Eval[1] and conduct user studies to directly assess improvements in visual effects. This approach allows for a clear and direct comparison of visual quality. We hope our exploration of material properties will inspire further interest and lead to the development of more comprehensive evaluation benchmarks.
>
>
> [1] GPT-4V(ision) is a Human-Aligned Evaluator for Text-to-3D Generation
>
>
> ***
>
> ### W3. Addressing Baked-in Shadows and Lighting Effects
>
> This issue is less pronounced in artist-created models, where albedo maps have less shadows and lighting effects. Our tests show that the albedo obtained from inverse rendering methods[1] closely matches the original artist-created albedo maps from the Objaverse dataset, as illustrated in the first row of Figure C3. For generative models, baked-in shading effects are more pronounced. Our method addresses this by integrating mature derendering algorithms, specifically the Intrinsic Anything[1], into our pipeline with minimal complexity. This integration derenders from four viewpoints and back-projects to obtain a better albedo map, then applies our Make-it-Real method for material painting. As shown in Figure C3's last row, this approach reduces lighting noise and supports a wider range of inputs. We will include more details about the derendering of inputs in the final version.
>
> [1] IntrinsicAnything: Learning Diffusion Priors for Inverse Rendering Under Unknown Illumination.

---

### Official Review · Reviewer_LVe6 · 2024-07-13

**Soundness:** 3
**Presentation:** 4
**Contribution:** 2
**Rating:** 5
**Confidence:** 4

**Summary:**

The proposed work leverages GPT-4V to extract and infer materials in albedo-only scenarios, utilizing existing material libraries to generate SVBRDF maps with a region-to-pixel algorithm. This approach enhances 3D mesh realism, ensures precise part-specific material matching, and is compatible with rendering engines, generating six comprehensive material maps. Developers need only paint albedo textures, with other material properties automatically generated, saving significant time. The contributions include using multimodal large language models for material recognition, creating a detailed material library, and developing a pipeline for high-quality material application to 3D assets.

**Strengths:**

- The method greatly enhances the realism of 3D objects by utilizing GPT-4V's extensive visual perception, supported by experimental evidence.
- It generates detailed PBR material maps (roughness, metallic, specular, normal, displacement, height) that are compatible with various rendering engines in a low-cost way.

**Weaknesses:**

- The method's effectiveness is largely dependent on the capabilities of MLLMs such as GPT-4V, which may not always produce accurate results. While the paper claims that the generated materials, represented as PBR maps, are compatible with downstream engines, it should be noted that this generation method does not rely on any physical principles of light transport and therefore cannot be genuinely considered as PBR maps.
- The method's performance might be constrained by the quality and detail of the initial albedo maps.

**Questions:**

- How are the PBR texture maps in the material library generated and how to ensure the accuracy of the material property?
- How does the proposed method perform with low-quality input albedo maps, such as those with many small UV pieces?

**Limitations:**

- The proposed method relies on GPT-4V for material generation from a generated database. However, many offline tools like ShaderMap can accomplish the same task more cost-effectively. These offline tools require no training and can produce comprehensive, high-quality material decompositions. Given that the generated materials are not physically-based, what advantages does the proposed method offer over efficient offline software that requires no training or fine-tuning?

---

> ### Author Rebuttal · Authors · 2024-08-05
>
> Thank you for your detailed review and insightful comments. We will address your questions and concerns below and in the revised paper.
>
> ***
>
> ### W1. Realism and Application of Generated PBR Maps
>
> Thank you for pointing this out. We acknowledge that the generated PBR maps are not true representations of physical maps, and we will clarify this in the final version paper. Nevertheless, our algorithm generates PBR maps that closely approximate the visual characteristics of true PBR properties. As demonstrated in Figure 5, our results significantly enhance the realism of 3D objects. Additionally, the analysis in the appendix (Figure 16) further examines the effects of each individual map, which mimic real textures, such as metallic properties (e.g., the dampening of the base albedo and increased surface shine on parts with high metallicity like scales and axes), roughness (with low roughness objects showing specular highlights), and displacement (fine bump effects on stone horses and bathtubs). In all cases, we produce visually comparable effects to true PBR maps.
>
> ***
>
> ### W2. Impact of Albedo Map Quality on Performance
>
> We agree that the final results can be affected by the quality and detail of the initial albedo maps. However, our method is designed to enhance the physical appearance even when the input albedo map varies in detail.
>
> On one hand, for objects that require meticulous craftsmanship and significant attention to detail, such as the truck and shield in Figure 5, our method precisely segments and assigns appropriate materials, significantly enhancing realism and producing high-quality objects with highly refined materials.
>
> On the other hand, for objects with relatively simple albedo maps that can be created quickly, like the saxophone shown in Figure C1, our method substantially improves the object's appearance under environmental lighting conditions, increasing realism and achieving a qualitative improvement.
>
> ***
>
> ### Q1. The clarification on PBR texture maps
>
> We would like to provide some clarification about this. The PBR texture maps in the material library are sourced from MatSynth. This dataset compiles high-quality PBR materials from publicly available resources like AmbientCG, which are widely used by artists for creative work. The materials are typically created using methods such as photogrammetry, scanning, and high-precision software, ensuring both visual fidelity and accurate physical properties.
>
> ***
>
> ### Q2. Performance with Fragmented UV Mapping
>
> Thanks for your insightful questions. Overly fragmented UV mapping degrades the performance of our method. As shown in the second column of  Figure C2, UV mappings with excessive fragmentation and color entanglement can cause the 2D segmented images to mix with other regions when reprojected into the UV space, leading to material blending issues. However, our method shows good results with original artist-created UV mappings and Blender's built-in mapping methods in Figure C2, such as `smart`, `sphere`, and `unwrap`. This indicates that our method still demonstrates good robustness with many mapping techniques.
>
> In practice, we observe that most objects in Objaverse have UV mappings with good properties, meaning they are not excessively fragmented into small pieces. Additionally, we can control the UV mapping process: For some low-quality UV maps and generative objects that originally lack UV maps, we can re-unwrap them to achieve higher quality.
>
> ***
>
> ### L1. Advantages of the Proposed Method Over Offline Tools
>
> Firstly, I would like to acknowledge that ShaderMap is a valuable tool for material creation, and it's great to see a thoughtful comparison of different approaches.
>
> Our method offers several distinct advantages:
>
> 1. **Focus on 3D Models**: Our approach excels in directly generating complete and fine-grained PBR maps for 3D models, offering more comprehensive integration with 3D assets compared to traditional tools. Tools like ShaderMap primarily focus on estimating higher-quality PBR texture maps from low-quality material images, such as photographs or instrument-captured material images, which may include lighting or shadows. These inputs are inherently material images and often do not focus on rendered views of 3D objects. In contrast, our method directly enhances 3D models, providing a more robust solution for integrating high-quality materials.
> 2. **Complexity and Usability**: Our method stands out for its ease of use and ability to produce semantically meaningful material assignments without requiring professional knowledge of 3D software. While ShaderMap can generate basic PBR maps (such as normal and specular maps) from albedo maps, this functionality currently yields relatively simple material effects. The material effects may appear to have lighting effects but do not necessarily match true semantic meanings. Since the entire albedo map is used as input, the software lacks fine-tuned regional specificity. Further manual adjustments are typically needed for local map values, requiring user interaction and expertise in 3D software, as well as additional time to achieve realistic results.

---

### Official Review · Reviewer_mcgT · 2024-07-13

**Soundness:** 3
**Presentation:** 3
**Contribution:** 3
**Rating:** 7
**Confidence:** 4

**Summary:**

This paper introduces the large-scale multimodal language models to realistic material rendering of 3D objects. Specifically, this paper employs MLLM to retrieve materials from a material library for different parts of . By combining 2D-3D alignment and diffuse reflection reference techniques, it generates and applies material texture maps to objects, achieving realistic rendering of 3D objects.

**Strengths:**

1. This paper introduces the MLLM into the texture inpainting pipeline, which is a interesting direction in the future.
2. The experiment results demonstrates the effectiveness of this method.
3. The paper is well-written and clearly articulated.

**Weaknesses:**

1. This paper shares some similarities with Mapa[1]. Could the authors further demonstrate the differences between Make-it-real and Mapa and the superiority of this method?
2. The necessity of MLLM in the pipeline requires further validation. How does the performance of the MLLM-based retrieval method compare to other retrieval methods? More experiments on the retrieval results are needed to show the superiority of the MLLM
-based methods.

[1]Zhang, Shangzhan, et al. "MaPa: Text-driven Photorealistic Material Painting for 3D Shapes." arXiv preprint arXiv:2404.17569 (2024).

**Questions:**

See the weakness.

**Limitations:**

The authors have not addressed the limitation in the paper.

---

> ### Author Rebuttal · Authors · 2024-08-06
>
> Thank you for your detailed review and insightful comments. We will address your questions and concerns below and in the revised paper.
>
> ***
>
> ### W1. Differences between Make-it-Real and MaPa and the superiority of our method.
>
> - **MaPa**: Introduces a text-driven segment-based procedural material graph representation. It uses a pre-trained 2D diffusion model as an intermediary bridge to connect text and material graphs. The method includes segment-controlled image generation and material graph optimization. It simply use CLIP for material retrieval.
> - **Make-it-Real**: Utilizes Multimodal Large Language Models (MLLMs), specifically GPT-4V, to recognize and apply real-world materials to 3D objects. The method involves texture segmentation, material matching, and SVBRDF map generation. It creates a comprehensive material library with detailed descriptions, allowing MLLMs to search and assign materials automatically.
>
> The core difference lies in how to perform the material retrieval. MaPa [1] still uses CLIP for the material retrieval process, while we rely directly on MLLM. We believe this core difference significantly improves the retrieval effectiveness for the following reasons:
>
> 1. CLIP can only focus on one area at a time. When generating images that need retrieval, other parts need to be masked while retaining the local part. This causes the retrieval process to lose the global semantic information of the object. MLLM, on the other hand, can retain the global semantic information by only circling the specified area in the pixels [2].
> 2. We creatively propose registering a hierarchical dictionary in MLLM by providing descriptions and images of material spheres, allowing MLLM to master a richer material library (The library is much larger and more practical with 1394 materials while MaPa only have 118).
> 3. With the rapid development of MLLM, transitioning this process from CLIP retrieval to a multi-step hierarchical and logical inference decision process relying on LLM is more promising with better explainability.
>
> To further validate the above points, we supplement the experiment detailed in W2 testing using MLLM for retrieval compared to using CLIP.
>
> There are also other differences that improving the performance compared to MaPa:
>
> 1. We propose a new method to achieve 3D segmentation leveraging Semantic-SAM and MLLM, which is more general than MaPa that adopt an out-of-date mesh segmentation method [3].
> ***
>
> ### W2. Experiments on the superiority of using MLLM for retrieval.
>
> we conducted additional experiments comparing GPT-4V (our method) and CLIP (MaPa method) to verify the superiority of using MLLM for retrieval.
>
> Specifically, we use the CLIP model as a baseline for comparison, which is the method adopted in PSDR-Scene [4] and MaPa [5]. We selected 100 objects from Objaverse with high object quality and annotated each object with different materials following the material classification of [4], resulting in 13 coarse types (Objaverse-C) and 80 subtypes (Objaverse-F). The retrieval results are reported in the table below. The results fully demonstrate the effectiveness of using MLLM for retrieval. We discover CLIP often fails in cases with metallic materials, largely due to the difficulty of extracting material regions without distortion. Our pipeline, using GPT-4V, queries four viewpoints for global shape and object semantics, can leveraging its strong open world prior knowledge to achieve promising results. Additionally, we observed that CLIP’s accuracy significantly drops when dealing with fine-grained categories compared to coarse categories, while GPT-4V maintains a high retrieval accuracy. This result also indicates that MLLM is more practical, as such systems often need to handle thousands of materials in a much finer granularity in real-world applications.
>
> |                | Objaverse-C | Objaverse-F |
> | :------------: | :---------: | :---------: |
> | GPT-4V / top-1 |  **70.59**  |  **64.71**  |
> | CLIP-L / top-1 |    28.53    |    10.89    |
> | CLIP-L / top-3 |    47.06    |    22.65    |
>
> ***
>
> [1] Zhang S, Peng S, Xu T, et al. MaPa: Text-driven Photorealistic Material Painting for 3D Shapes
>
> [2] Yang J, Zhang H, Li F, et al. Set-of-mark prompting unleashes extraordinary visual grounding in gpt-4v
>
> [3] Sagi Katz and Ayell, et al. Hierarchical mesh decomposition using fuzzy clustering and cuts
>
> [4] Yan K, Luan F, Hašan M, et al. Psdr-room: Single photo to scene using differentiable rendering
>
> [5] Vecchio G, Deschaintre V. MatSynth: A Modern PBR Materials Dataset

---

### Official Review · Reviewer_s3Ca · 2024-07-14

**Soundness:** 4
**Presentation:** 4
**Contribution:** 4
**Rating:** 7
**Confidence:** 4

**Summary:**

This paper tackles the problem of recovering materials for 3D meshes with known geometry and base color. The proposed solution consists of three steps:

- Utilizing Semantic-SAM to segment the 3D objects, identifying and isolating various material regions.
- Using hierarchical prompting to retrieve and match materials from a material library.
- Generating the actual SVBRDF map based on the albedo and the retrieved materials.

**Strengths:**

- The qualitative results look promising for both artist-created assets and objects generated by 3D generative models.

- The paper explores and demonstrates the potential of leveraging large multimodal language models to describe material properties of the objects—an exciting finding that could inspire future research.

- The design of hierarchical prompting for material library retrieval presents an intuitive and effective way to improve the efficiency of the query.

**Weaknesses:**

- In the user study, participants are asked to compare the base mesh and the refined mesh. However, users may naturally prefer objects with enhanced lighting effects, which does not effectively evaluate the quality of the generated materials. A more informative approach might be to compare an artist-created mesh with material to the same shape with a re-predicted material, which could provide more insights into material quality.

- For the application presented in this paper, a common issue is that 3D models with a single albedo map often have baked-in shadows and other lighting effects. This is common in both artist-created models and especially in outputs from 3D generative models (e.g. is the shadow in Fig 7 bottom baked in albedo?). It's unclear how the proposed method addresses or might even be misled by these artifacts since it assumes that similar colors represent similar BRDF values in estimation. This could be a limitation when compared to alternative methods that directly optimize or predict the texture map. A discussion on this could be helpful to clarify.

- The discussion on performance metrics is missing, such as runtime and the average number of parts and queries per object.

**Questions:**

- How are viewpoints selected?

- Does the method depend on good UV mapping?

- How are the categories within the material library constructed?

**Limitations:**

The authors discuss the limitations of their method and broader impact in the paper.

---

> ### Author Rebuttal · Authors · 2024-08-05
>
> Thanks for your detailed review and insightful comments! We will address your questions below and in the revised paper.
> ***
> ### W1: More Comparisons with Artist-Created Materials:
> Thank you for the insightful suggestion! We conducted an additional user study comparing materials generated by our Make-it-Real method to those created by artists. We filtered the Objaverse dataset for objects with material maps and randomly sampled 200 objects for the experiment.
>
> Our results are shown in the following table. The second row highlights our method's superiority over albedo-only objects, with 73.1% of users recognizing enhanced material effects, reinforcing the paper's conclusions. The first row indicates that users rated the material effects generated by our method as superior to or on par with artist-created materials for 61.6% of the objects. This is a highly promising result.
> |                               | Win/Tie Combined% | A/B Win% | A/B Tie% | A/B Lose% |
> | :---------------------------: | :---------------: | :------: | :------: | :-------: |
> |    Make-it-Real v.s. Human    |       61.6        |   30.5   |   31.1   |   38.4    |
> | Make-it-Real v.s. Albedo only |         88.2         |   73.1   |   15.1   |   11.8    |
>
> We examined these cases closely, as visualized in Figure C1. For objects like pistol, wooden chair, speakers, and saxophone, Make-it-Real showed strong similarity to artist-made materials, maintaining consistent metallicity, roughness, highlights, and coloration. In cases such as shield, boots, landline phone, and oil drum, our method produced materials that, while different, are realistic and sometimes visually superior. Our pipeline approaches the level of refinement seen in some artist-created materials, which often require significant time and effort, and clearly outperforms more basic and crude artist-generated materials. This highlights its effectiveness and potential for practical applications.
>
> ***
>
> ### W2: Addressing Baked-in Shadows and Lighting Effects
>
> This issue is less pronounced in artist-created models, where albedo maps have less shadows and lighting effects. Our tests show that the albedo obtained from inverse rendering methods[1] closely matches the original artist-created albedo maps from the Objaverse dataset, as illustrated in the first row of Figure C3. For generative models, baked-in shading effects are more pronounced. Our method addresses this by integrating mature derendering algorithms, specifically the Intrinsic Anything[1], into our pipeline with minimal complexity. This integration derenders from four viewpoints and back-projects to obtain a better albedo map, then applies our Make-it-Real method for material painting. As shown in Figure C3's last row, this approach reduces lighting noise and supports a wider range of inputs. We will include more details about the derendering of inputs in the final version.
>
> [1] IntrinsicAnything: Learning Diffusion Priors for Inverse Rendering Under Unknown Illumination.
>
> ***
>
> ### W3: Results of Performance Metrics
>
> Thanks for your valuable feedback. We have added tests for average performance metrics, as shown in the table below.
>
> | Run Time(min) | # Material Parts/obj. | # Queris/obj. | Memory Cost(GB) |
> | :-----------: | :-------------------: | :-----------: | :-------------: |
> |     1.54      |         2.42          |     5.76      |      12.52      |
>
>
> Additionally, we conducted further tests to measure the time required for each of the three stages drawn in our pipeline: (a) Rendering and Segmentation, (b) Material Retrieval, and (c) Material Generation.
>
>
> |                   | Render & Seg | Mat Retrieval | Mat Genertaion | Total |
> | :---------------: | :--------------------: | :---------------------: | :---------------------: | :---: |
> | **Run Time(min)** |          0.26          |          0.49           |          0.79           | 1.54  |
>
> ***
>
> ### Q1. The clarification of viewpoint selection
>
> We explain some of the detalis in Sec 3.2.1, and here we provide further details. Initially, we perform rasterization for rendering, and the selection of viewpoints follows the approach used in TEXTure. We render a total of ten viewpoints: eight at 45-degree intervals around the horizontal axis and two additional views, one top-down and one bottom-up, forming our initial set of rendered images.
>
> Among these rendered images, we select the viewpoint with the largest foreground area (the projected area in rasterization) as our primary viewpoint because it is more likely to contain more material information. This viewpoint is then used for subsequent material retrieval and map generation. We further refine the material generation using our proposed material inpainting method in Sec 3.2.3.
>
> ***
>
> ### Q2. Discussion of UV mapping of input mesh
>
> We conducted experiments to evaluate the impact of UV mapping on our method. As shown in Figure C2, except for the second column where the lighting UV mapping method results in fragmented and interwoven colors, other unwrapping methods such as smart, sphere, and unwrap produce better results. Our method does not perform well with overly fragmented UV maps but demonstrates good robustness with other mapping methods.
>
> In practice, we have observed that most objects in Objaverse have UV mappings with good properties, meaning they are not overly fragmented into small pieces. Additionally, for some low-quality UV maps and generative objects that originally lack UV maps, we can  control UV mapping, (re-)unwrapping them to achieve higher quality.
>
> ***
>
> ### Q3. Categories construction in material library
>
> These materials originally come from AmbientCG, an open online repository, where each material has a fine-grained category name, such as "Planks". We follow the classification of material categories in MatSynth, resulting in 13 major categories and 80 subcategories. This forms our two-level hierarchical tree structure, which corresponds to our hierarchical prompt retrieval.

---

### Author Rebuttal · Authors · 2024-08-07

We thank all reviewers and appreciate the constructive comments and the recognition of novelty, and we are grateful for that most reviewers score our work with positive initial ratings (two accept, and two borderline accept). Our work introduces the MLLM into the texture inpainting pipeline **[mcgT]**, which is an exciting finding **[s3Ca]** and a interesting direction in the future **[s3Ca, mcgT]**. The plug-and-play **[dDLL]** design with accelerated hierarchical prompting **[jx7i]** can achieve SVBRDF map generation in a low-cost way **[LVe6]**.

Regarding the weaknesses and questions proposed by the reviewers, we have provided detailed responses to each reviewer separately. Here, we summarize the key questions and general points that we believe will interest all reviewers.

***

### Summary of Key Questions/Comments

**S1.** Reviewer s3Ca, dDLL, jx7i point out that Baked lighting effects on albedo map do harm to quality.

**S2.** Reviewer LVe6, s3Ca about Performance with Fragmented UV Mapping.

**S3.** Reviewer s3Ca about Comparison with artists created materials.

***

We will then address the comments/questions summarized above by referring to the new experimental results attached in the single-page PDF:

**GR1. (for S1) Baked lighting effects [Figure C3]**

This issue is less pronounced in artist-created models, where albedo maps have less shadows and lighting effects. Our tests show that the albedo obtained from inverse rendering methods[1] closely matches the original artist-created albedo maps from the Objaverse dataset, as illustrated in the first row of Figure C3. For generative models, baked-in shading effects are more pronounced. Our method addresses this by integrating mature derendering algorithms, specifically the Intrinsic Anything[1], into our pipeline with minimal complexity. This integration derenders from four viewpoints and back-projects to obtain a better albedo map, then applies our Make-it-Real method for material painting. As shown in Figure C3's last row, this approach reduces lighting noise and supports a wider range of inputs. We will include more details about the derendering of inputs in the final version.


***

**GR2. (for S2) Performance with different UV mapping [Figure C2]**

Overly fragmented UV mapping degrades the performance of our method. As shown in the second column of  Figure C2, UV mappings with excessive fragmentation and color entanglement can cause the 2D segmented images to mix with other regions when reprojected into the UV space, leading to material blending issues. However, our method shows good results with original artist-created UV mappings and Blender's built-in mapping methods in Figure C2, such as smart (default setting in blender), sphere, and unwrap. This indicates that our method still demonstrates good robustness with many mapping techniques.

In practice, we observe that most objects in Objaverse have UV mappings with good properties, meaning they are not excessively fragmented into small pieces. Additionally, we can control the UV mapping process: For some low-quality UV maps and generative objects that originally lack UV maps, we can re-unwrap them to achieve higher quality.

***

**GR3. (for S3) More Comparisons with Artist-Created Materials [Figure C1]**

We visualize some of the results comparing our Make-it-real generated materials with artists. Quantative results are detailed in rebuttal response to Reviewer s3Ca.

***

We sincerely thank the reviewers for their insightful comments and suggestions on our work. We will incorporate all of the rebuttal experiments and discussions into the revised version of our work.

***

[1] Chen, Xi, et al. IntrinsicAnything: Learning Diffusion Priors for Inverse Rendering Under Unknown Illumination.

[2] Boss M, Huang Z, Vasishta A, et al. SF3D: Stable Fast 3D Mesh Reconstruction with UV-unwrapping and Illumination Disentanglement.

---

### Decision · Program_Chairs · 2024-09-25

**Decision:**

Accept (poster)

**Comment:**

After reading the rebuttal, four out of five reviewers supported acceptance. One reviewer remained negative insisting that the assumptions about the input (meshes with given albedo maps) are simplistic and that most of the contribution comes from the use of the GPT-4V. The AC does not agree that the assumption is a problem -- there are already methods that can populate meshes with initial albedo maps (e.g., Text2Tex) and basically they solve an orthogonal problem. The technical novelty is indeed not that impressive wrt material retrieval (i.e., prompting GPT-4V for material retrieval), yet: (a) the idea of leveraging large multimodal language models to describe material properties is still underexplored and can inspire future research (as mentioned by s3Ca), (b) the segmentation module of Fig. 2 and optimization of Eq. 3 are novel enough components. The AC agrees with the rest of the reviewers that the contribution of the paper is above the bar, and the results deserve recognition. Thus, the AC recommends acceptance and strongly encourages the authors to include material from the rebuttal in the final version of the paper.